# Large-scale forward genetics screening identifies Trpa1 as a chemosensor for predator odor-evoked innate fear behaviors

Yibing Wang[1,2], Liqin Cao[3], Chia-Ying Lee[3], Tomohiko Matsuo[4], Kejia Wu[1,2], Greg Asher[3], Lijun Tang[4], Tsuyoshi Saitoh [3], Jamie Russell[5], Daniela Klewe-Nebenius[3], Li Wang[2], Shingo Soya[3], Emi Hasegawa[3], Yoan Chérasse [3], Jiamin Zhou[2], Yuwenbin Li[2], Tao Wang [5], Xiaowei Zhan[5], Chika Miyoshi[3], Yoko Irukayama[3], Jie Cao[6], Julian P. Meeks [6], Laurent Gautron[7], Zhiqiang Wang[3], Katsuyasu Sakurai[3], Hiromasa Funato [3,8], Takeshi Sakurai[3], Masashi Yanagisawa [3,9], Hiroshi Nagase[3], Reiko Kobayakawa[4], Ko Kobayakawa[4], Bruce Beutler[5] & Qinghua Liu[1,2,3,5,6,10]

Innate behaviors are genetically encoded, but their underlying molecular mechanisms remain largely unknown. Predator odor 2,4,5-trimethyl-3-thiazoline (TMT) and its potent analog 2-methyl-2-thiazoline (2MT) are believed to activate specific odorant receptors to elicit innate fear/defensive behaviors in naive mice. Here, we conduct a large-scale recessive genetics screen of ethylnitrosourea (ENU)-mutagenized mice. We find that loss of Trpa1, a pungency/irritancy receptor, diminishes TMT/2MT and snake skin-evoked innate fear/defensive responses. Accordingly, $Trpa1^{-/-}$ mice fail to effectively activate known fear/stress brain centers upon 2MT exposure, despite their apparent ability to smell and learn to fear 2MT. Moreover, Trpa1 acts as a chemosensor for 2MT/TMT and Trpa1-expressing trigeminal ganglion neurons contribute critically to 2MT-evoked freezing. Our results indicate that Trpa1-mediated nociception plays a crucial role in predator odor-evoked innate fear/defensive behaviors. The work establishes the first forward genetics screen to uncover the molecular mechanism of innate fear, a basic emotion and evolutionarily conserved survival mechanism.

[1] National Institute of Biological Sciences, 102206 Beijing, China. [2] Department of Biochemistry, University of Texas Southwestern Medical Center, Dallas, TX 75390, USA. [3] International Institute for Integrative Sleep Medicine (WPI-IIIS), University of Tsukuba, Tsukuba, Ibaraki 305-8575, Japan. [4] Functional Neuroscience Lab, Kansai Medical University, Hirakata, Osaka 573-1010, Japan. [5] Center for Genetics of Host Defense, University of Texas Southwestern Medical Center, Dallas, TX 75390, USA. [6] Department of Neuroscience, University of Texas Southwestern Medical Center, Dallas, TX 75390, USA. [7] Department of Internal Medicine, University of Texas Southwestern Medical Center, Dallas, TX 75390, USA. [8] Department of Anatomy, Faculty of Medicine, Toho University, Ota-Ku, Tokyo 143-8540, Japan. [9] Life Science Center, Tsukuba Advanced Research Alliance, University of Tsukuba, Tsukuba, Ibaraki 305-8575, Japan. [10] Tsinghua Institute of Multidisciplinary Biomedical Research, Tsinghua University, 100084 Beijing, China. These authors contributed equally: Yibing Wang, Liqin Cao, Chia-Ying Lee. Correspondence and requests for materials should be addressed to K.K. (email: skobayak@me.com) or to B.B. (email: bruce.beutler@utsouthwestern.edu) or to Q.L. (email: qinghua.liu@utsouthwestern.edu)

Innate behaviors are prevalent and advantageous for survival in animal world. An innate behavior has at least three important traits: (1) it is instinctive and does not require learning; even an animal raised in isolation can perform the behavior when exposed to the stimulus for the first time; (2) it is stereotypic; all individuals of the same species will do it the same way every time; (3) it is heritable, passing from generation to generation through genes. However, the genetic bases of innate behaviors are largely unknown.

Fear is a basic emotion that triggers characteristic defensive behaviors and physiological responses to promote survival in dangerous situations[1–3]. Fear can be induced by both innate and learned mechanisms. In Pavlovian fear conditioning paradigms, rodents are trained to freeze in response to a conditioned (e.g., olfactory, auditory, or visual) stimulus by pairing it with an unconditioned stimulus (e.g., electric footshock). On the other hand, laboratory rodents are instinctively afraid of snakes or cats despite being isolated from predators for generations. This innate fear represents an evolutionarily conserved and genetically encoded pro-survival mechanism[4]. Therefore, we hypothesized that it might be feasible to investigate the molecular mechanism of innate fear using a forward genetics approach.

Predator odors or kairomones, from the fur, skin, urine, feces, and excretory glands of the predators, have been widely used to evoke innate fear/defensive behaviors, such as avoidance and risk assessment behaviors, in laboratory mice[5,6]. One of the best-studied kairomones is fox odorant 2,4,5-trimethyl-3-thiazoline (TMT), which can also induce instinctive freezing in naive mice[6]. A potent analog of TMT, 2-methyl-2-thiazoline (2MT), has recently been shown to elicit highly robust freezing response in mice[7]. It is widely believed that specific odorant receptors are responsible for sensing TMT/2MT to elicit innate fear/defensive behaviors[8–12]. Accordingly, genetic lesion of dorsal class II olfactory sensory neurons (OSNs) rendered mice defective for TMT-induced avoidance[9]. At least two dozens of candidate odorant receptors for TMT have been identified[8,13]. Notably, optogenetic activation of Olfr1019-expressing OSNs induces immobility, whereas Olfr1019 knockout mice are partially defective for TMT-induced freezing[13]. Moreover, other TMT sensing mechanisms seem to exist because the Grueneberg ganglion (GG), which does not typically express odorant receptors, have been implicated in TMT-evoked freezing[14,15]. It has also been proposed that the trigeminal system plays a critical role in TMT-evoked freezing behavior[6,16,17].

The transient receptor potential ankyrin 1 (Trpa1), a member of the TRP family of ion channels, has been shown to function as a chemical, mechanical, and temperature sensor[18–20]. In particular, Trpa1 is a well-known chemoreceptor for pungent natural compounds, such as cinnamaldehyde, allyl isothiocyanate and mustard oil[21–23], as well as harmful environmental irritants, such as formalin and acrolein (tear gas)[24–26]. Accordingly, Trpa1 is highly expressed in the somatosensory systems, such as the trigeminal ganglia (TG), dorsal root ganglia and nodose ganglia, and has important functions in nociception, i.e., sensing harmful and potentially painful stimuli[27,28]. Here, we performed a large-scale recessive genetics screen of randomly mutagenized mice using a highly robust 2MT-evoked innate fear assay. We found that the Trpa1-mediated chemosensory system, in addition to classical olfactory system, plays a central role in mediating predator odor-evoked innate fear/defensive behaviors in mice.

## Results

### Development of a highly robust 2MT-evoked innate fear assay.
A highly robust behavioral assay is essential for setting up a forward genetics screen to identify the genes required for innate fear. Previous attempts of forward genetics screening on learned fear have not been successful because these behavioral assays are too variable, with a relative standard deviation (RSD) ranging 50–100%[29]. By contrast, innate fear behaviors are simpler and less variable than learned fear behaviors. Unlike TMT, 2MT is not noxious, is sold cheaply, and elicits robust freezing behavior in mice[7]. Thus, we selected 2MT to develop a simple and highly robust innate fear assay suitable for high-throughput mouse screening (Fig. 1, and Supplementary Fig. 1). By optimizing the age of mice (>15 week), 2MT dose (10 μl) and assay duration (15 min) (Fig. 1a–d), we showed that wild-type C57BL/6J male mice exhibited 0–15% freezing rate without odor and 55–95% freezing rate in response to 2MT exposure (Fig. 1a). This optimized assay had a RSD value of ~10% (Supplementary Fig. 1e–g), making it feasible to screen for "fearless" mutant mice that show <47% freezing rate (i.e., >3 SD below the normal mean of 77% freezing rate) upon 2MT exposure.

### Identification of fearless mutant mice by genetic screening.
We designed a recessive mouse screening platform for instant association of phenotypes with causative mutations induced by the chemical mutagen ENU in C57BL/6J background[30]. Each ENU-mutagenized G1 founder male was crossed with wild-type females to generate G2 daughters, which were mated back with the G1 male to create a family of 15–50 G3 mice (Fig. 2a). We performed exome sequencing on every G1 founder and genotyped all G2 and G3 mice by ion torrent sequencing. The average G3 mouse was homozygous for $5.68 \pm 3.8$ mutations; heterozygous for $24.7 \pm 10.8$ mutations; and indeterminate genotype for $1.01 \pm 4.2$ mutations (standard deviation shown). After phenotypic screening of G3 mice, this unique strategy allowed for immediate identification of the causative mutation by direct linkage analysis of the ENU-induced random point mutations with the phenotype of interest[30].

We screened 13,222 G3 mice (males and females, from 632 pedigrees) for abnormal freezing response upon 2MT exposure. In all, 33,069 coding/splicing mutations in 13,828 genes were examined. Approximately 6.5% of all genes were destroyed or damaged sufficiently to cause a phenotype, with the mutant alleles tested three times or more in the homozygous state[31]. Among the mutant families that we identified, one mutant pedigree named fearless (frl) contained four phenovariants that showed 20–30% freezing rates upon 2MT exposure (Fig. 2b and Supplementary Movie 1). Manhattan plot indicated that this "fearless" phenotype was strongly linked ($P = 2.13 \times 10^{-11}$) to a homozygous mutation in the Trpa1 gene (Fig. 2c, d and Supplementary Fig. 2a). This 5′ splice site mutation likely created a null allele of Trpa1 encoding a mis-spliced and unstable mRNA that specified an inactive Trpa1 protein lacking all of the transmembrane domains (Fig. 2e–h). By contrast, we also screened multiple Trp mutant families, none of which showed abnormal freezing response (Supplementary Fig. 3). These results suggest that Trpa1 plays a crucial role in mediating 2MT-evoked innate fear responses.

### Trpa1⁻/⁻ mice are deficient for innate fear responses.
To confirm that this ENU-induced Trpa1 mutation was the sole cause of the observed fearless phenotype, we examined an independently constructed Trpa1 knockout strain[32], which removed exons 22–24 encoding the essential pore-forming domain of Trpa1 channel (Supplementary Fig. 2b). Previous studies have shown that homozygous $Trpa1^{-/-}$ mice are healthy and fertile except for some behavioral deficits to irritant and proalgesic agents[24,32]. We found that $Trpa1^{-/-}$ mice exhibited normal anxiety behaviors in the open field test and elevated-plus maze test (Supplementary Fig. 4a, b). However, $Trpa1^{-/-}$ mice

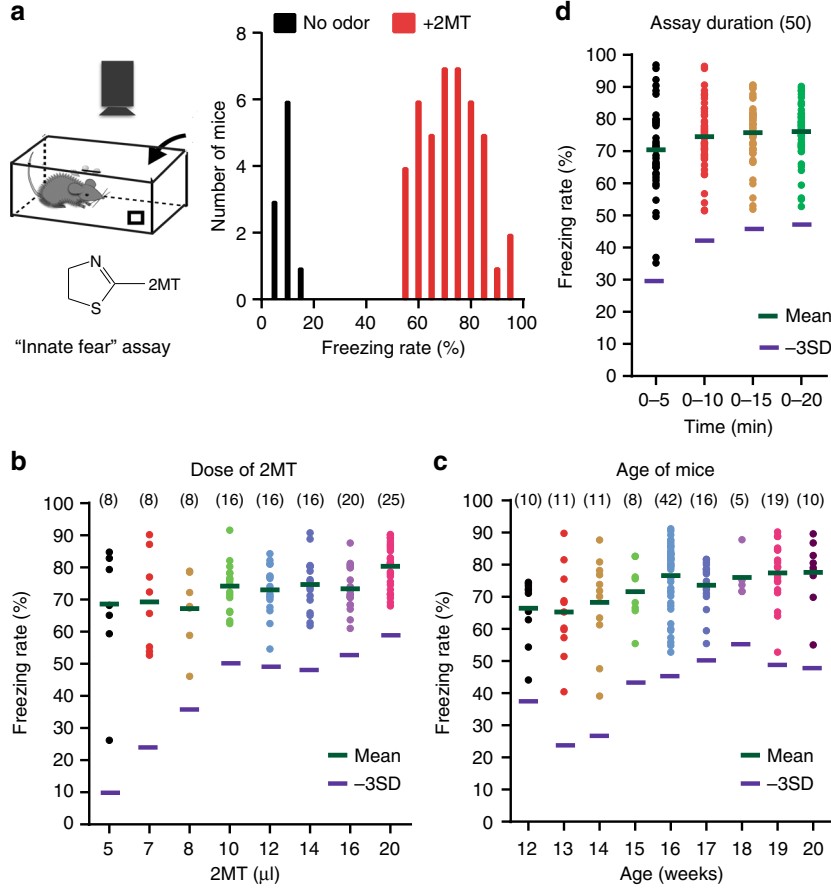

**Fig. 1** Development of a highly robust innate fear assay. **a** A schematic of 2MT-evoked innate fear assay (left). A quantitative graph showing the distribution of freezing rates of C57BL/6J males with or without (20 μl = $2.1 \times 10^{-4}$ mole) 2MT exposure (right). **b** A graph showing the distribution of freezing rates (average 20 min) in C57BL/6J males (>15 week old) at different doses of 2MT exposure. **c** A graph showing the distribution of freezing rates (average 20 min) at different ages of C57BL/6J males exposed to 20 μl of 2MT. **d** A graph showing the distribution of freezing rates for different assay duration using C57BL/6J males. −3SD, 3 standard deviation (SD) below the normal mean of freezing rates upon 2MT exposure. The numbers in parentheses indicate the sample numbers for individual conditions

displayed diminished freezing response to 2MT or TMT as compared to wild-type and heterozygous littermates (Fig. 3a, Supplementary Fig. 4c, and Supplementary Movie 2). Similarly, 2-*sec*-butyl-4,5-dihydrothiazole (SBT), a 2MT/TMT-related alarm pheromone[15], evoked reduced freezing response in *Trpa1*$^{-/-}$ mice than in *Trpa1*$^{+/-}$ mice (Supplementary Fig. 4d). Because the majority of Trpa1$^+$ neurons co-express Trpv1[28,33], we also examined *Trpv1* knockout mice[34] to investigate whether Trpv1 functioned in tandem with Trpa1 in this innate fear response (Supplementary Fig. 2c). Unlike *Trpa1*$^{-/-}$ mice, *Trpv1*$^{-/-}$ mice exhibited normal freezing response to 2MT/TMT (Fig. 3b), suggesting that Trpv1 was dispensable for 2MT/TMT-evoked innate freezing. Finally, 2MT exposure resulted in a significant surge in the plasma level of stress hormone corticosterone in *Trpa1*$^{+/-}$ mice, and this physiological response was almost abolished in *Trpa1*$^{-/-}$ mice (Fig. 3c).

We exposed *Trpa1* knockout mice to low concentration (1/100 of the dose used for screening) of 2MT to further investigate the role of Trpa1 in other innate fear/defensive behaviors. As expected, *Trpa1*$^{-/-}$ mice displayed a significant deficit in low-dose 2MT-evoked freezing, avoidance, flight, and risk assessment behaviors as compared to *Trpa1*$^{+/-}$ littermates (Fig. 3d, e and Supplementary Movie 3)[1,35]. Furthermore, we used snake skin molt as a source of natural predator odors to elicit innate fear/defensive behaviors in *Trpa1*$^{+/-}$ and *Trpa1*$^{-/-}$ mice. Accordingly, *Trpa1*$^{-/-}$ mice spent much less time than *Trpa1*$^{+/-}$ mice

in freezing (average 3.6 vs. 34.1%) and avoidance (average 3.9 vs. 29.6%), and exhibited less flight (average 0.6 vs. 8 times) and risk assessment (average 2.4 vs. 23.5 times) behaviors (Fig. 3f, g and Supplementary Movie 4). On the other hand, *Trpa1*$^{-/-}$ mice spent significant more time than *Trpa1*$^{+/-}$ littermates in investigation (average 60.2 vs. 1%), including sniffing, moving, biting, and tearing the snake skin (Fig. 3f, g and Supplementary Movie 4). These results suggest that Trpa1 plays an essential role in predator odor-evoked innate fear/defensive behaviors.

***Trpa1*$^{-/-}$ mice can smell and learn to fear 2MT/TMT.** It has previously been shown that 2MT exposure induces the immediate early gene *c-fos* expression in the central nucleus of amygdala (CeA), which communicates with downstream brain regions, such as the ventral periaqueductal gray (vPAG), to mediate innate and learned freezing behaviors[7,36]. Accordingly, we observed that 2MT-evoked *c-fos* induction was greatly diminished in the CeA, vPAG, and the paraventricular nucleus (PVN) of hypothalamus in *Trpa1*$^{-/-}$ brains relative to *Trpa1*$^{+/-}$ brains (Fig. 4a, b). The PVN is frequently activated in fearful or stressful situations, as part of the hypothalamic–pituitary–adrenal axis, to stimulate stress hormone production[37]. Therefore, Trpa1 is required for activation of known fear/stress centers in the mouse brain upon 2MT exposure.

It is plausible that Trpa1 is involved in olfaction because modest Trpa1 expression has been detected in the olfactory

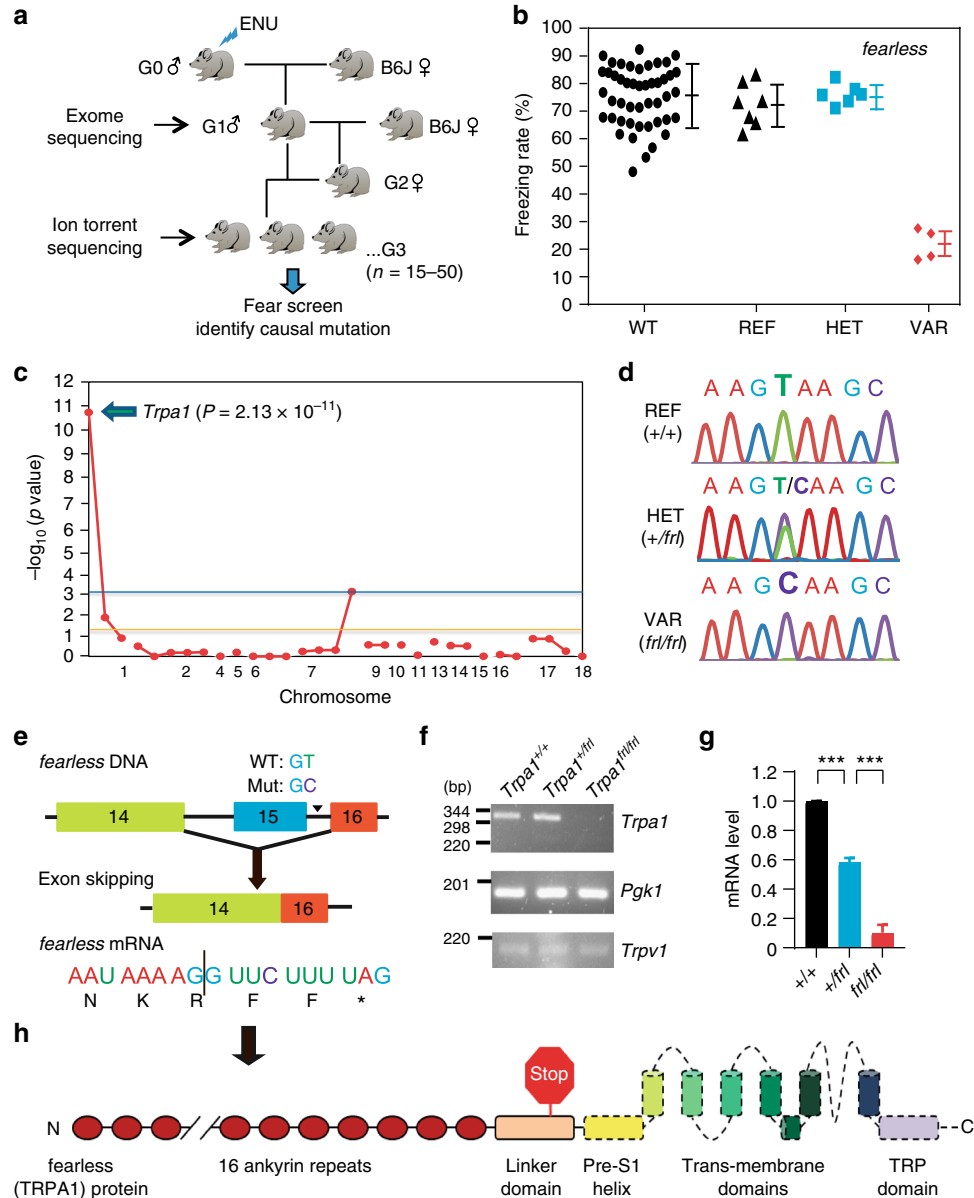

**Fig. 2** Identification of *fearless* (*Trpa1*) mutant mice by forward genetics screening. **a** A flowchart of the recessive fear screen. **b** A graph of the *fearless* mutant pedigree consisting of seven reference (REF), six heterozygous (HET), and four phenovariant (VAR) individuals. A separate group of wild-type (WT) mice were included as controls every week. **c** A Manhattan plot showing a strong linkage ($P = 2.13 \times 10^{-11}$) between the *Trpa1* mutation and fearless phenotype. Horizontal yellow and blue lines represent the thresholds of $P < 0.05$ without or with Bonferroni correction, respectively. **d** The causative mutation was identified by sequencing the *Trpa1* gene of REF, HET, and VAR mice. **e** A schematic of partial exon/intron structure of *Trpa1* gene. The 5' splice site (GT to GC) mutation results in the skipping of exon 15 and introduction of a premature stop codon. **f, g** Semi-quantitative (**f**) and quantitative (**g**) RT-PCR analysis showing that *fearless* mutant *Trpa1* mRNA is diminished possibly owing to nonsense-mediated decay. Data are presented as mean ± SEM ($n = 4$, Student's *t*-test, \*\*\*$P < 0.001$). **h** A schematic of the domain structure of Trpa1 protein, with the stop sign marking the premature stop codon and dash line representing the domains missing in mutant Trpa1 protein

epithelium[27,38]. However, we found that $Trpa1^{-/-}$ mice were not anosmic because they were as efficient as $Trpa1^{+/+}$ and $Trpa1^{+/-}$ littermates in finding the hidden food pellet (Supplementary Fig. 5a). Moreover, we observed that 2MT induced equivalent levels of *c-fos* expression in the olfactory bulbs (OB) of $Trpa1^{+/-}$ and $Trpa1^{-/-}$ mice, including the dorsal region of OB that was specifically activated by TMT[11,15] (Fig. 4a, b and Supplementary Fig. 5b–d). It has been reported that a subset of olfactory-activated neurons from the cortical amygdala (CoA) are important for TMT-induced avoidance behavior[39]. However, 2MT exposure induced a similar low level of *c-fos* expression in the CoA of $Trpa1^{+/-}$ and $Trpa1^{-/-}$ mice (Fig. 4a, b).

Furthermore, habituation-dishabituation test showed that $Trpa1^{-/-}$ mice exhibited the same level of detection threshold for 2MT as $Trpa1^{+/-}$ mice (Fig. 4c). Homozygous *Trpa1* mutant mice were proficient in dual odor-based and sound-based fear conditioning assays (Fig. 4d and Supplementary Fig. 6), suggesting that they could distinguish different odors and exhibited normal learned fear responses. Importantly, $Trpa1^{-/-}$ mice could be further trained to fear 2MT by pairing it with electric footshocks (Fig. 4e). Taken together, these findings suggest that $Trpa1^{-/-}$ mice appear to have a normal sense of smell and can learn to fear 2MT.

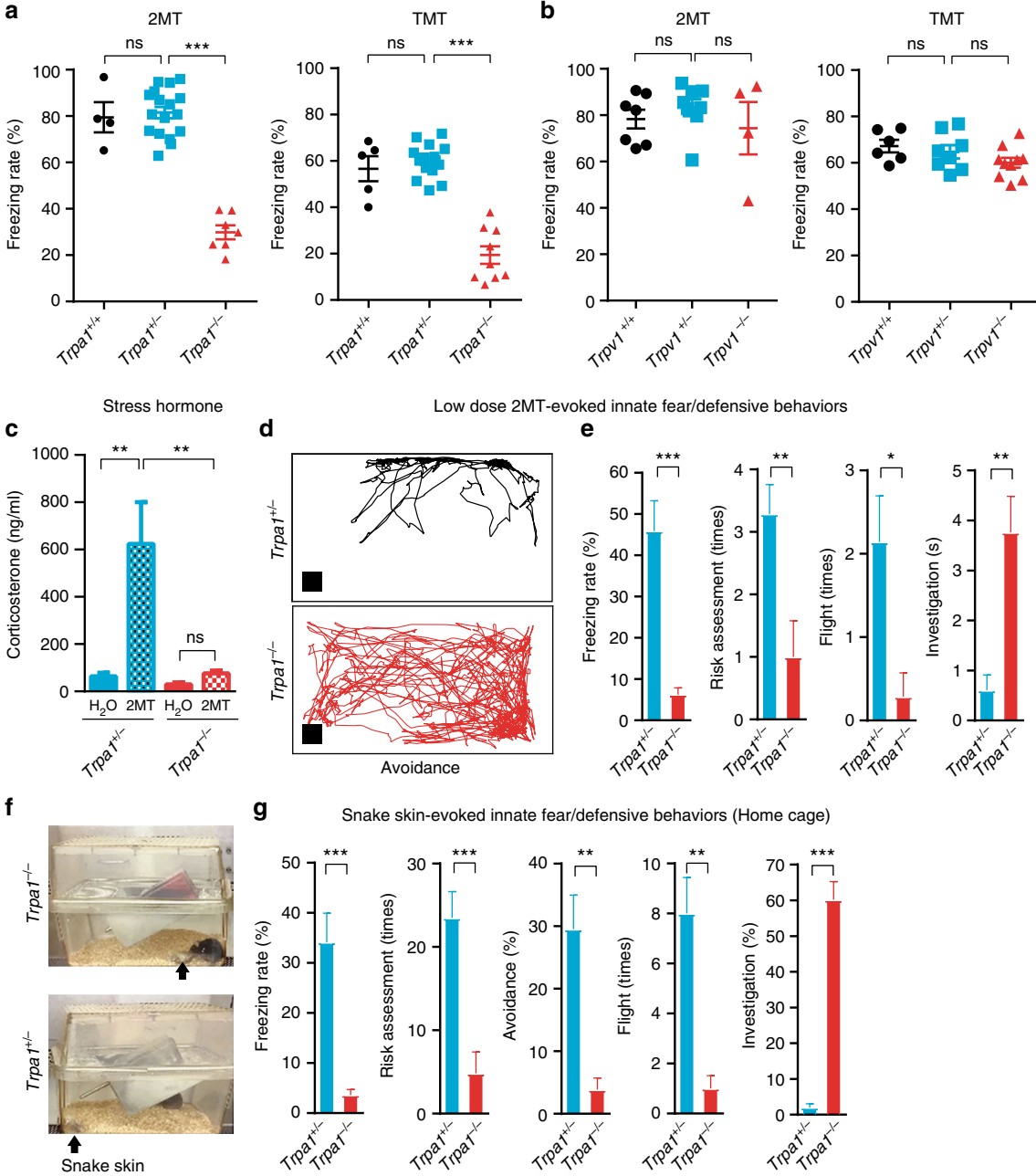

**Fig. 3** Trpa1 mediates 2MT/TMT/snake skin-evoked innate fear/defensive responses. **a**, **b** Wild-type, heterozygous, and homozygous *Trpa1* (**a**) and *Trpv1* (**b**) knockout mice were examined for innate freezing behaviors evoked by 2MT or TMT. Data are presented as mean ± SEM (Student's *t*-test, ***P < 0.001; **P < 0.01; ns not significant). **c** Plasma concentration of stress hormone corticosterone was measured in *Trpa1+/−* and *Trpa1−/−* mice following exposure to $H_2O$ or 2MT for 20 min. Data are presented as mean ± SEM ($n = 6$, two-way ANOVA test, **P < 0.01). **d** Representative avoidance tracks of *Trpa1+/−* and *Trpa1−/−* mice in response to low concentration ($1.05 \times 10^{-6}$ mole) of 2MT exposure in the test cage. The black box indicates the position of 2MT-containing filter paper. **e** Quantitative analysis of freezing, risk assessment, flight, and investigation behaviors of *Trpa1+/−* and *Trpa1−/−* mice in testing cages in response to low-dose 2MT exposure. Data are presented as mean ± SEM ($n = 7$, Student's *t*-test, ***P < 0.001; **P < 0.01, *P < 0.05). **f** An image depicting the snake skin-evoked innate fear assay in the home cage. **g** Quantitative analysis of snake skin-evoked freezing, risk assessment, avoidance, flight, and investigation behaviors of *Trpa1+/−* and *Trpa1−/−* mice. Data are presented as mean ± SEM ($n = 6$, Student's *t*-test, ***P < 0.001; **P < 0.01)

**Trpa1 acts as a chemosensor for TMT/2MT in HEK293T cells.** Because TMT is a pungent odor, we hypothesized that Trpa1, a well-known pungency/irritancy receptor[21–26], function as a chemosensor for TMT-like thiazolines. To test this hypothesis, we performed $Ca^{2+}$ imaging in HEK293T cells that were transiently transfected with the mCherry or Trpa1-P2A-mCherry constructs (Supplementary Fig. 7a). Both TMT and 2MT, but not 2-methyl-2-oxazoline (2MO), a structurally related non-fear inducing

odorant (Supplementary Fig. 1b), evoked $Ca^{2+}$ transients in the Trpa1-expressing cells in a dose-dependent manner (Fig. 5a). Inhibition of Trpa1 by HC-030031 treatment[26] abolished 2MT-evoked $Ca^{2+}$ transients in transfected HEK293T cells (Supplementary Fig. 7b). Furthermore, we compared 12 mammalian Trp channels that belong to all four subfamilies, Trpa, Trpc, Trpm, and Trpv[18–20], for 2MT/TMT-evoked $Ca^{2+}$ responses. Only Trpa1, but not other Trp channels, could be specifically activated

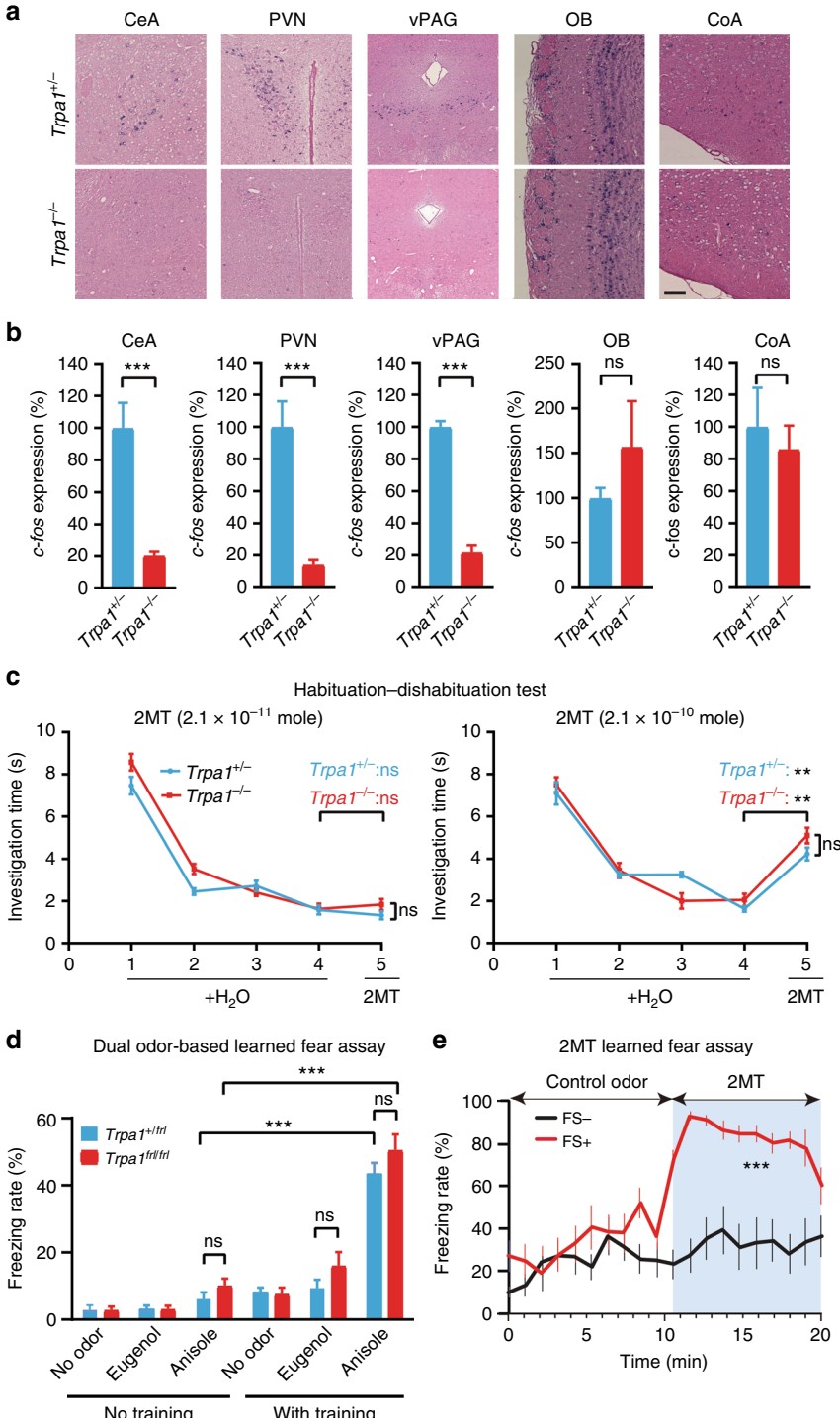

**Fig. 4** *Trpa1$^{-/-}$* mice can smell and learn to fear 2MT. **a** Representative images showing *c-fos* mRNA in situ hybridization (ISH) of the central nucleus of amygdala (CeA), paraventricular nucleus (PVN) of hypothalamus, ventral periaqueductal gray (vPAG), olfactory bulb (OB), and cortical amygdala (CoA) regions of the brains of 2MT-exposed *Trpa1$^{+/-}$* and *Trpa1$^{-/-}$* mice (bar: 100 μm). **b** Quantitative analysis of *c-fos*-positive neurons in the CeA, PVN, vPAG, OB, and CoA of *Trpa1$^{+/-}$* and *Trpa1$^{-/-}$* mouse brains (**a**). The relative measure (%) of *c-fos* signals of *Trpa1$^{-/-}$* samples is normalized to those of *Trpa1$^{+/-}$* controls. Data are presented as mean ± SEM ($n = 4$, Student's *t*-test, ***$P < 0.001$; ns not significant). **c** Habituation-dishabituation test results of *Trpa1$^{+/-}$* and *Trpa1$^{-/-}$* mice exposed to filter papers containing H$_2$O (trials 1–4) or 2MT (trial 5). 2MT dose was shown above the left and right graphs (left graph: $n = 6$ for both *Trpa1$^{+/-}$* and *Trpa1$^{-/-}$* mice, Student's *t*-test; ns not significant; right graph: $n = 5$ for *Trpa1$^{+/-}$* mice and $n = 4$ for *Trpa1$^{-/-}$* mice, **$P < 0.01$). **d** Dual odor-based learned fear assay was performed with *Trpa1$^{+/frl}$* and *Trpa1$^{frl/frl}$* mutant mice by pairing anisole with electric footshock and using eugenol as a negative control. Data are presented as mean ± SEM ($n = 10$, Student's *t*-test, ***$P < 0.001$; ns not significant). **e** *Trpa1$^{-/-}$* mice were untrained (−FS) or trained (+FS) to fear 2MT by pairing 2MT exposure with electric footshocks. Eugenol was used as a control odor. Data are presented as mean ± SEM ($n = 6$, Student's *t*-test, ***$P < 0.001$; **$P < 0.01$; *$P < 0.05$)

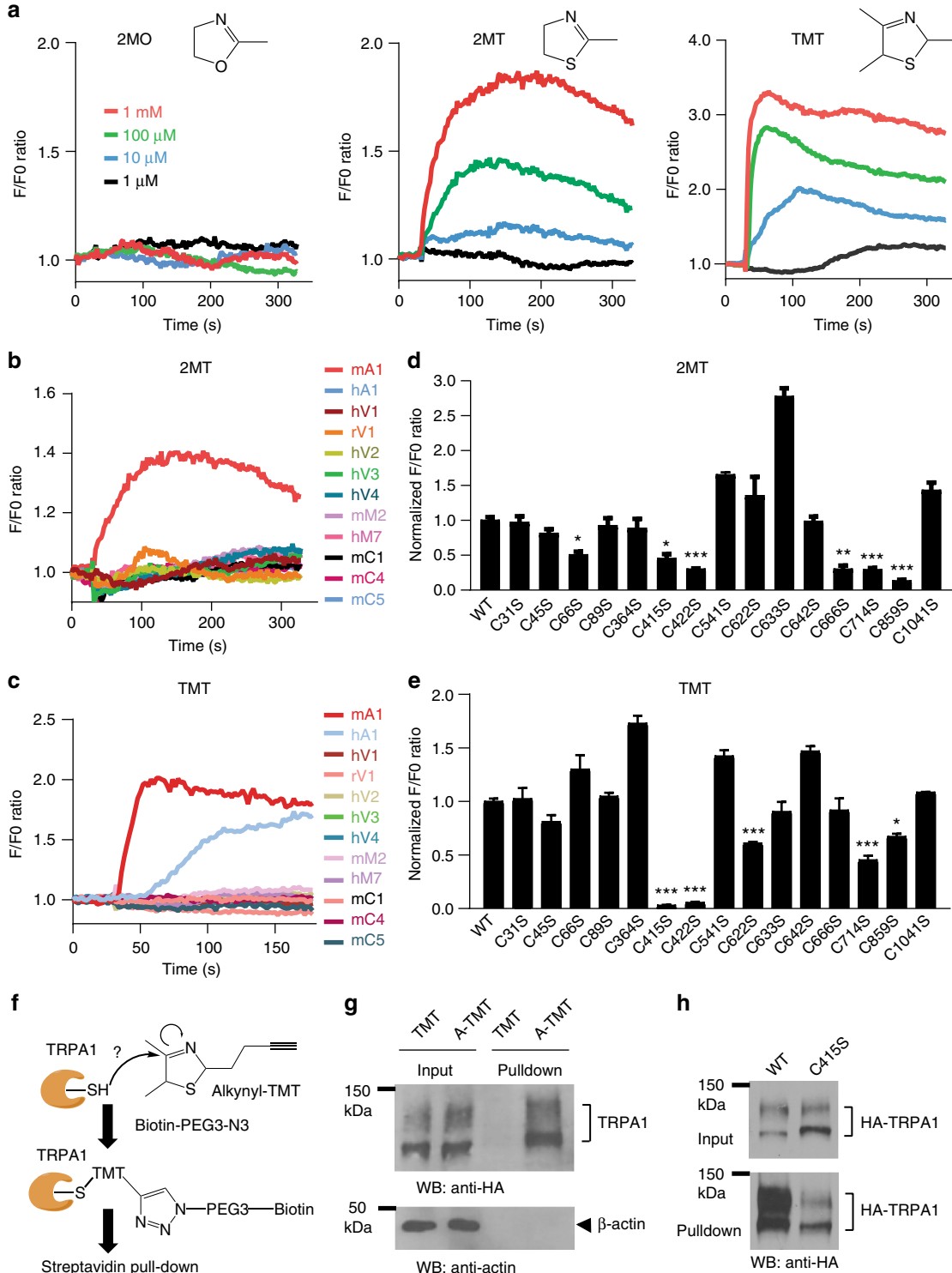

**Fig. 5** Trpa1 acts as a chemosensor for 2MT/TMT. **a** Twenty-four hours after transfection of the Trpa1-P2A-mCherry construct, $Ca^{2+}$ imaging was performed in HEK293T cells upon exposure to 1, 10, 100, 1000 μM of 2MO, 2MT, or TMT. **b**, **c** 2MT (100 μM, **b**) or TMT (50 μM, **c**)-evoked $Ca^{2+}$ response curves of HEK293T cells after co-transfection of the mCherry plasmid with each of 12 constructs expressing different mammalian TRP channels (h human, r rat, m mouse). **d**, **e** Quantitative analysis of wild-type (WT) and mutant Trpa1 activities in response to 2MT (100 μM, **d**) or TMT (50 μM, **e**) by $Ca^{2+}$ imaging in transfected HEK293T cells. Y-axis, F/F0 ratio normalized to WT construct. Data are presented as mean ± SEM ($n = 3$ biological replicates). Pairwise comparisons were made for each mutant Trpa1 to wild-type Trpa1 using two-way unpaired Student's t-test. Multiple comparisons were controlled by Bonferroni correction with unadjusted P-value cutoff being 0.005 (***), 0.01 (**), and 0.05 (*). **f** A schematic of in vivo labeling of transfected HA-Trpa1 with alkynyl-TMT followed by biotin conjugation via click chemistry and streptavidin pull down. **g**, **h** Immunoblots showing that alkynyl-TMT (A-TMT), but not TMT, could pull down HA-Trpa1 (**g**), and that alkynyl-TMT could pull down C415S mutant less efficiently than wild-type HA-Trpa1 (**h**) from transfected HEK293T cells

by 2MT/TMT (Fig. 5b, c and Supplementary Fig. 7c–e). Notably, TMT modestly activated, whereas 2MT failed to activate human TRPA1 in HEK293T cells (Fig. 5b, c), suggesting species-specific differences in Trpa1-mediated $Ca^{2+}$ responses. These results suggest that mouse Trpa1 could function as a sensitive chemosensor for TMT/2MT in heterologous cells.

Multiple pungent chemicals as well as oxygen could directly activate human TRPA1 channel by covalent modification of critical cysteine residues within its cytoplasmic domain[25,40,41]. To determine whether 2MT/TMT activated Trpa1 via a similar mechanism, we substituted a number of cysteines with serines in mouse Trpa1 protein, and compared the activities of wild-type and mutant Trpa1 by $Ca^{2+}$ imaging in transfected HEK293T cells. We showed that substitution of Cys66, Cys415, Cys422, Cys622, Cys666, Cys714, or Cys859 with Ser abolished or reduced 2MT/TMT-evoked $Ca^{2+}$ responses (Fig. 5d, e). By contrast, Icilin, a TRPA1 ligand that did not react with cysteine[25], activated wild-type and mutant Trpa1 to a similar extent (Supplementary Fig. 7f, g).

To further study the activation mechanism, we synthesized alkynyl-TMT by adding an alkynyl moiety to the C2 position of TMT (Supplementary Fig. 8). If alkynyl-TMT could covalently label HA-TRPA1 ectopically expressed in HEK293T cells, we should be able to precipitate modified Trpa1 proteins by biotin conjugation via click chemistry and streptavidin pulldown (Fig. 5f)[25]. Accordingly, our results showed that alkynyl-TMT-treated, but not TMT-treated, Trpa1 was precipitated from transfected HEK293T cells by streptavidin beads (Fig. 5g). Moreover, the C415S mutant Trpa1 was precipitated less

efficiently than wild-type Trpa1 from alkynyl-TMT-treated HEK293T cells (Fig. 5h). These results, coupled with site-directed mutagenesis studies, suggest that Trpa1 may act as a direct chemosensor for TMT/2MT through covalent modifications of critical Cys residues of the receptor.

**Trpa1 is essential for 2MT sensing by TG neurons.** Trpa1 is highly expressed in the TG that directly innervate the nasal cavity. Thus, we performed c-Fos immunohistochemistry (IHC) to examine whether 2MT activated the Trpa1-expressing TG neurons in vivo. Notably, 2MT exposure elicited strong nuclear c-Fos signals in specific TG neurons of heterozygous, but not homozygous, Trpa1 mutant mice (Fig. 6a, b). Consistent with that TG neurons project into the caudal subnucleus of spinal trigeminal nucleus (Sp5C)[42,43], we also found that 2MT exposure induced c-fos mRNA expression in the Sp5C region of $Trpa1^{+/-}$, but not $Trpa1^{-/-}$, mouse brains (Fig. 6c, d). Double Trpa1/c-Fos staining revealed that on average 67% of Trpa1-expressing TG neurons showed nuclear c-Fos signals following 2MT exposure (Fig. 6e, f and Supplementary Fig. 9). Approximately half of c-Fos-positive TG neurons lacked Trpa1 expression (Fig. 6e, f), but their activation in response to 2MT was also abolished in Trpa1-deficient mice (Fig. 6a, b). The $Fos^+/Trpa1^-$ TG neurons could be the result of intraganglion neural transmission as previously reported[44,45]. Specifically, 2MT-activated $Trpa1^+$ primary sensory neurons could secrete neuromodulators to stimulate downstream $Trpa1^-$ neurons in the TG. These observations suggest that Trpa1 is absolutely required for 2MT sensing by a subset of TG neurons.

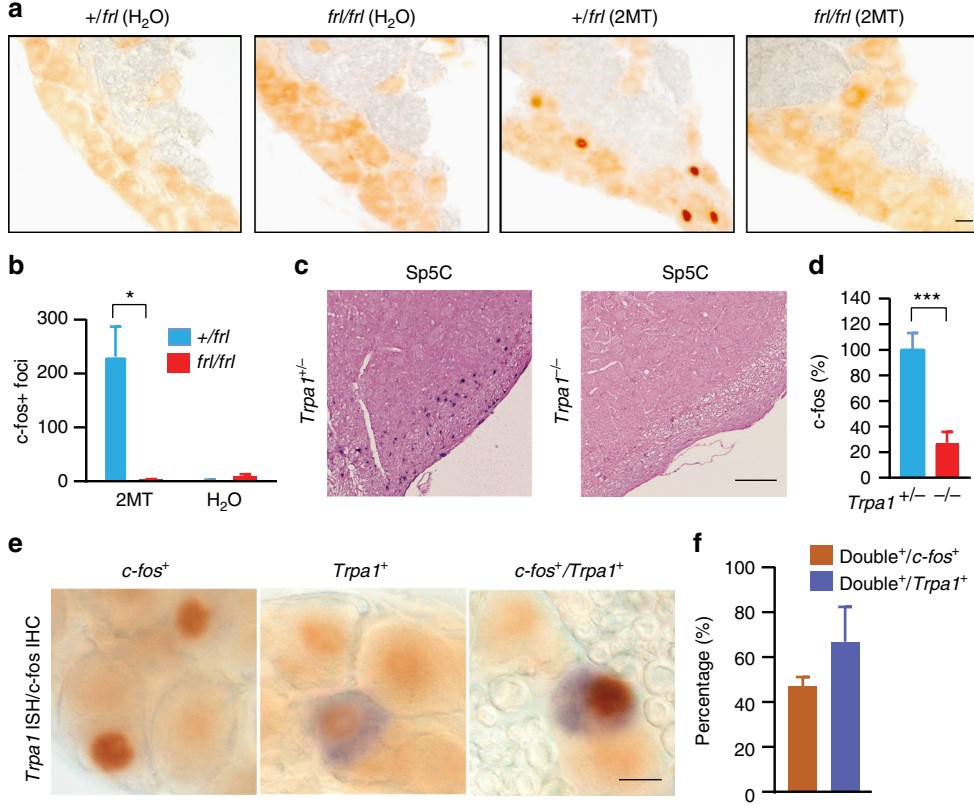

**Fig. 6** Trpa1 is essential for 2MT sensing by a subset of TG neurons. **a**, **b** Immunohistochemistry (IHC) (**a**) and quantitation (**b**) of c-Fos-expressing TG neurons of $Trpa1^{+/frl}$ and $Trpa1^{frl/frl}$ mice after $H_2O$ or 2MT exposure. Data are presented as mean ± SEM ($n = 4$, Student's t-test, *$P < 0.05$). **c**, **d** In situ hybridization (ISH) (**a**) and quantitation (**b**) of c-fos-expressing neurons in the Sp5C of $Trpa1^{+/-}$ and $Trpa1^{-/-}$ mice after 2MT exposure. The relative measure (%) of c-fos signals of $Trpa1^{-/-}$ samples is normalized to those of $Trpa1^{+/-}$ samples (**d**). Data are presented as mean ± SEM ($n = 4$, Student's t-test, ***$P < 0.001$). **e**, **f** Double staining (**e**) and quantitation (**f**) of $Trpa1^+$, $c\text{-}Fos^+$ double positive TG neurons of $Trpa1^{+/frl}$ mice after 2MT exposure. Scale bars are 100 μm in **a** and 10 μm in **c** and **e**

**Trpa1$^+$ TG neurons are critical for 2MT-evoked freezing**. We attempted classical lesion experiments to examine the contribution of the trigeminal system to 2MT-evoked innate freezing behavior. However, most of the bilateral TG lesioned (blTGx) mice died within a couple of days after the operation. Thus, we performed unilateral TG lesion (ulTGx) instead in wild-type mice (Fig. 7a). In ulTGx mice, 2MT-evoked c-fos expression in the Sp5C region was abolished in the lesion side as compared to the control side, confirming that TG neurons acted upstream of Sp5C neurons (Fig. 7b, c). Although we did not expect to see a behavioral phenotype, unilateral loss of the trigeminal input was sufficient to significantly attenuate 2MT-evoked innate freezing behavior (Fig. 7d, e).

To further evaluate the functional importance of Trpa1$^+$ TG neurons, we bilaterally injected adeno-associated virus expressing Trpa1 (AAV-Trpa1) into the TG of Trpa1$^{-/-}$ mice (Fig. 7f). Mice injected with AAV-GFP bilaterally were used as negative controls (Fig. 7g). Ectopic expression of Trpa1 in the TG could significantly rescue 2MT-evoked c-Fos activation in TG neurons of Trpa1$^{-/-}$ mice (Fig. 7h, i). Moreover, the majority of c-Fos-expressing TG neurons in response to 2MT were those that ectopically expressed Trpa1 in Trpa1$^{-/-}$ mice (Fig. 7h, i). Furthermore, AAV-Trpa1 rescued Trpa1$^{-/-}$ mice showed significantly enhanced (by 2-fold) 2MT-evoked innate freezing as compared to AAV-GFP-injected mice (Fig. 7j, k). Taken together, these classical lesions and AAV rescue experiments strongly suggest that Trpa1-expressing TG neurons contribute critically to 2MT-evoked innate freezing behavior.

## Discussion

In this study, we developed a highly robust 2MT-evoked innate fear assay and established a forward genetics screen to identify randomly mutagenized mice with abnormal fear responses. This unbiased fear screen identified that inactivation of Trpa1, a pungency/irritancy receptor, diminished 2MT/TMT and snake skin-evoked innate fear/defensive behaviors. Accordingly, 2MT exposure failed to efficiently activate known fear/stress centers in the brain of Trpa1$^{-/-}$ mice, despite their apparent ability to smell and learn to fear 2MT. Furthermore, Trpa1 could function as a chemosensor for 2MT/TMT and was absolutely essential for 2MT sensing by TG neurons. We showed that Trpa1-expressing TG neurons contributed critically to 2MT-evoked innate freezing behavior. These results suggest that the Trpa1-dependent somatosensory system plays a prominent role in mediating predator odor-evoked innate fear/defensive responses.

The innate ability to detect chemical warning signals from predators is absolutely essential for species survival. We are just beginning to understand the molecular mechanisms by which preys detect a variety of predator scents. The trace amine-associated receptor 4 (Taar4), acts as the primary receptor responsible for elicitation of avoidance behavior in response to 2-phenylethylamine, isolated from carnivore urines[46]. Trpc2 in the vomeronasal organ plays a crucial role in the detection of non-volatile major urinary proteins from rats or cats to elicit avoidance and risk assessment behaviors[47]. Here, we identified Trpa1 as a potential novel chemosensor that mediates thiazolines (TMT/2MT/SBT) and snake skin-evoked innate freezing and other fear/defensive behaviors. Alternatively, it is possible that Trpa1 deficiency causes wiring/developmental defects in TG. However, this possibility is unlikely because TG injection of AAV-Trpa1 rescued 2MT-evoked innate freezing behaviors. Moreover, in contrast to Trpc2 that is expressed in almost all vomeronasal neurons, Trpa1 is expressed in only a small subset (~10%) of TG neurons, and thus cannot function as a housekeeping molecule for overall TG functions. Collectively, these studies suggest that

mice use novel chemoreceptors as well as conventional odorant receptors to detect potentially dangerous chemicals in the environment, including kairomones from predators, to induce innate fear/defensive responses.

It has long been a subject of debate whether TMT is a bona fide predator odor or simply a pungent odor, and whether the olfactory or trigeminal system mediates TMT-elicited innate fear/defensive behaviors[6,16,17,48]. Through unbiased forward genetics screening, we identified that Trpa1 deficiency resulted in diminished 2MT/TMT-evoked innate freezing responses. Whereas unilateral TG lesion significantly attenuated 2MT-evoked freezing in wild-type mice, bilateral TG injection of AAV-Trpa1 partially rescued 2MT-evoked innate freezing in Trpa1$^{-/-}$ mice. Our results, coupled with previous studies[8–13,39], establish that both the trigeminal and olfactory systems play important roles in 2MT/TMT-evoked innate fear/defensive behaviors. It should be noted that Trpa1 might function in multiple sensory systems. We cannot exclude the possibility that an unidentified subset of Trpa1-expressing olfactory neurons may play a partial role in the predator odor-evoked innate fear/defensive responses. For example, Trpa1 may act as a chemosensor in the GG neurons that have been implicated in TMT-evoked innate freezing[14,15]. Future studies are needed to identify other types of tissues/neurons where Trpa1 functions and delineate the neural circuitries that underlie thiazolines-evoked innate fear/defensive responses.

In almost all of learned fear assays, animals are trained to fear a neutral stimulus by pairing it with an aversive stimulus, such as electric footshock or LiCl injection that causes pain or discomfort[49–51]. Thus, this type of learned fear is actually built on a pain-induced innate fear circuit[52,53]. In principle, predator odor-mediated activation of the Trpa1 nociceptive pathway should instinctively warn the mice of imminent dangers and trigger emergency responses to promote survival. Our studies provide a compelling molecular logic to explain why predator odor-evoked innate fear/defensive behaviors are genetically hardwired.

Forward genetics screening in mice represents one of the most effective approaches to understanding the genetic basis of human physiology and disease[54]. Emotions define the essence of being human and are instinctive drivers of behaviors[2,3]; and yet little is understood about the molecular bases of emotions. Although rapid progress has been made in mapping the neural circuitries of innate and learned fear[10,12,55–58], the molecular circuits underlying fear, particularly the core signaling pathways that mediate various fear/defensive responses, remain poorly understood. Our results substantiate the utility of a forward genetics approach to investigate the molecular mechanism of innate fear and perhaps other innate behavioral paradigms.

## Methods

**Mice and genotyping**. Animal protocols used in this study were approved by the International Institute for Integrative Sleep Medicine at University of Tsukuba, Japan, as well as by the Institutional Animal Care and Use Committee at UT Southwestern Medical Center (UTSWMC) at Dallas, USA. Mice were housed in groups of 4–5 and maintained on a 12-h light-dark schedule (lights on at 06:00) with ad libitum access to food and water. C57BL/6J mice were purchased from Jackson Laboratory (JAX, Bar Harbor ME), and mutagenized[59] and bred with two pedigree structures at UTSWMC as previously described[30]. Mice were at least 14 weeks old at the start of fear screening. B6;129P-Trpa1$^{tm1Kykw}$/J (stock number 006401, herein referred to Trpa1$^{-/-}$) and B6.129X1-Trpv1$^{tm1jul}$/J mice (stock number 003770, herein referred to Trpv1$^{-/-}$) mice were purchased from JAX. It should be noted that although Trpa1$^{-/-}$ and Trpv1$^{-/-}$ mice have a mixed genetic background, the phenotypes of Trpa1$^{-/-}$ mice are consistent with those of homozygous Trpa1 ENU mutant mice in C57BL/6J background. All experiments were repeated independently for at least two times. We do not employ any method of randomization to determine how samples/animals were allocated. For all animal experiments, we chose at least $n = 6$ for each sample group to ensure adequate statistical power. The experimenters were blind to the genotypes of mice when performing the experiments and analyzing the results.

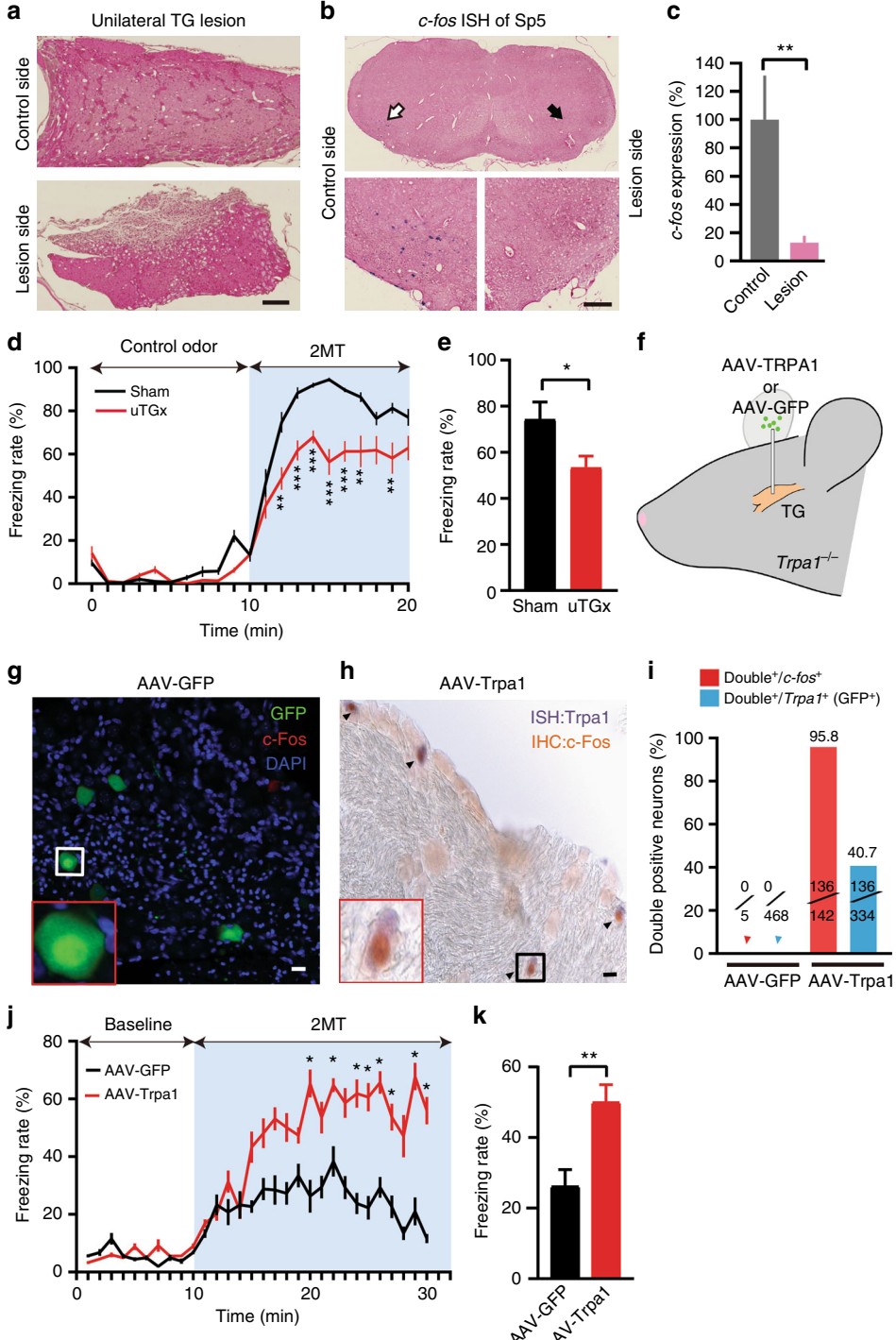

**Fig. 7** Trpa1[+] TG neurons contribute critically to 2MT-evoked innate freezing. **a** Hematoxylin and eosin staining of the control and lesioned TG of wild-type mouse after unilateral lesion (ulTGx). **b** Representative images showing *c-fos* ISH signals in the Sp5C regions of ulTGx mice after 2MT exposure. Enlarged images of the control (white arrow) and lesion sides (black arrow) are shown in the lower panels. Scale bars are 100 μm in **a** and **b**. **c** Quantitative analysis of *c-fos* ISH signals in the Sp5C regions of the control and lesion sides of ulTGx mice after 2MT exposure. The relative measure (%) of *c-fos* signals of the lesion side is normalized to that of control side. Data are presented as mean ± SEM (*n* = 8, Student's *t*-test, **P < 0.01). **d** Quantitative analysis of 2MT-evoked freezing behavior in the sham (black) and ulTGx (red) mice. Data are presented as mean ± SEM (*n* = 8, Student's *t*-test, **P < 0.01, ***P < 0.001). **e** Quantification of average freezing rate in the sham (black) and ulTGx (red) mice during 10 min of 2MT treatment. Data are presented as mean ± SEM (*n* = 8, Student's *t*-test, *P < 0.05). **f** A schematic of bilateral injection of AAV-GFP or AAV-Trpa1 into the TG of *Trpa1*[−/−] mice. **g** Representative IHC images of AAV-GFP-infected TG sections of *Trpa1*[−/−] mice. GFP (green); c-Fos (red); DAPI (blue). **h** Representative images of double *Trpa1*/c-Fos staining of AAV-Trpa1-infected TG sections of *Trpa1*[−/−] mice. *Trpa1* ISH (purple); c-Fos IHC (brown). Arrowheads indicate *Trpa1*[+]/c-Fos[+] TG neurons. Insets are high magnification images. Scale bar is 20 μm. **i** Quantitation of 2MT-evoked GFP[+], c-Fos[+], or *Trpa1*[+]/c-Fos[+] TG double positive neurons in AAV-GFP or AAV-Trpa1-infected TG, respectively. **j**, **k** Quantitative analysis of 2MT-evoked freezing behavior in the AAV-GFP (blue) and AAV-Trpa1 (red)-injected mice. Data are presented as mean ± SEM (*n* = 6, Student's *t*-test, *P < 0.05; **P < 0.01)

For mouse genotyping, mouse tails were cut and lysed in 100 µl All[ele]-In-One Mouse Tail Direct Lysis Buffer (Cat# ABP-PP-MT01500, Allele Biotechnology, San Diego, CA, USA) at 55 °C overnight. The lysate was heated at 90 °C for 10 min to inactivate proteinase, cooled down, and centrifuged at $12,000 \times g$ for 2 min. Genotyping of $Trpa1$ and $Trpv1$ knockout mice were carried out by PCR using the primers designed by JAX. For genotyping the $Trpa1$ fearless ($frl$) ENU mutant allele, the point mutation of $Trpa1$ gene was detected using the derived cleaved and amplified polymorphic sequence (dCAPS) method. Briefly, genomic DNA was amplified using 5′-TGGTAGAATACCTCCCCGAGTGCATGAATG-3′ and 5′-TGTGAAGAGCATTCATTCAGC-3′, which produced a BsmI restriction enzyme recognition site specific for the mutant $Trpa1$ allele. The PCR products were digested with BsmI and resolved by 3.5% agarose gel electrophoresis.

**Development of a highly robust 2MT-evoked innate fear assay.** Odorants 2MT and 2MO were purchased from Tokyo Chemical Industry (Tokyo, Japan), whereas TMT was initially purchased from Contech (Waterford, CT) and later chemically synthesized using the synthetic scheme described below. For comparison of innate freezing behavior in response to different odors, a constant concentration of test odorant-containing gas (10 ppm vol/vol) was produced using a calibration gas generator (Permeator PD-1B-2; Gastec, Kanagawa, Japan) and pumped into a sealed test cage ($31.5 \times 19.5 \times 13$ cm) via a tygon tube. A volume of odorant gas equal to that of the test cage was pumped into the test cage prior to the introduction of each mouse. Odor presentation was performed in a chemical fume hood. Mouse behaviors were recorded with a digital video camera (7.5 frames/s, $320 \times 240$ pixels) connected to a computer running the FreezeFrame software (Actimetrics, Wilmette, IL, USA). The videos were analyzed by FreezeFrame to calculate the freezing rate[7]. A mouse was considered to be freezing if no movement was detected for ≥2 s. We used C57BL/6J male mice to systematically optimize the 2MT-evoked innate fear assay by varying the mouse age, 2MT dose and assay duration. Moreover, we used C57BL/6J males exclusively for the majority of behavioral experiments. However, we assayed both male and female mice from the ENU G3 mutant pedigrees during forward genetics screening. In general, mice of at least 14 weeks of age were individually introduced into an open test cage ($19.1 \times 29.1 \times 12.7$ cm, Ancare Corporation, Cat# N10HT). A small ($2 \times 2$ cm) filter paper containing 10 µl ($1.05 \times 10^{-4}$ mole) 2MT, 35 µl ($2.98 \times 10^{-4}$ mole) TMT, or 50 µl SBT ($3.49 \times 10^{-4}$ mole) was placed at a corner of test cage for 15–20 min. Mouse freezing behavior was analyzed as described above.

**Forward genetic screening and automated genetic mapping.** G3 mice were pre-genotyped at all mutation sites using ampliseq panels and Ion Torrent sequencing. Mice were evaluated by pedigree for phenotypic anomalies using the 2MT-evoked innate fear assay alongside wild-type controls. Raw data (freezing rate) were entered into the Linkage Analyzer program, which tested the statistical association between assay performance and genotype for all mice at every mutation site in every pedigree, using dominant, additive, and recessive models of inheritance. The data were visualized in the form of a Manhattan plot in Linkage Explorer ($-\log(P)$ vs. chromosomal location).

**Quantitative RT-PCR analysis of *Trpa1* expression.** TG were dissected from $Trpa1^{+/+}$, $Trpa1^{+/frl}$, $Trpa1^{frl/frl}$ mice and total RNA was extracted by Trizol (Thermo Fisher Scientific, USA). Total cDNA was synthesized from 1 µg RNA using the High Capacity cDNA Reverse Transcription kit (Thermo Fisher Scientific, USA), according to manufacturer's protocol. For qPCR, cDNA template was mixed with $2 \times$ iTaq™ universal SYBR® Green supermix (Biorad, CA, USA) and primers in 384-well plate. The qPCR experiments were performed on C1000 Touch™ Thermal Cycler and CFX384 Real-Time System (Biorad). The amplification curves as well as Ct value were analyzed by Bio-rad CFX Manager. All the assays were performed in quadruplicate and ΔCt value was calculated compared to $Pgk1$. The results were processed and plotted by Prism 7.0 (GraphPad, CA, USA). The following primers were used: $Trpa1$-F: 5′-TGCTGACATCCTCCTGAACA-3′ and $Trpa1$-R: 5′-ACATCCTGGGTAGGTGCTACT-3′; $Trpv1$-F: 5′-CGAG-GATGGGAAGAATAACTCACTG-3′ and $Trpv1$-R: 5′-GGATGATGAAGA-CAGCCTTGAAGTC-3′; $Pgk1$-F: 5′-AACCTCCGCTTTCATGTAGAG-3′ and $Pgk1$-R: 5′-GACATCTCCTAGTTTGGACAGTG-3′. We observed a faint band below the predicted PCR product in $Trpa1^{+/frl}$ and $Trpa1^{frl/frl}$ samples in Fig. 2f. Large-scale PCR amplification and direct sequencing of this lower band showed that the 5′ splice site mutation of $Trpa1$ caused skipping of exon 15, resulting in a frame-shift and introduction of a premature stop codon as shown in Fig. 2h.

**Analysis of low-dose 2MT and snake skin-evoked innate fear and defensive behaviors.** $Trpa1^{+/-}$ and $Trpa1^{-/-}$ mice were individually habituated in a new and open test cage (Ancare N10HT $19.1 \times 29.1 \times 12.7$ cm) for 15 min. We excluded the mice showing persistent immobility during habituation for analysis. After habituation, a small filter paper laced with low dose ($1.05 \times 10^{-6}$ mole) 2MT was presented at a corner of the cage for 5 min. Mouse behaviors were recorded with a digital video camera (7.5 frames/s, $320 \times 240$ pixels) from the top and with an iPhone from the side. The following fear-like and defensive behaviors are measured: The videos from the digital camera were analyzed to generate (1) freezing rate by the FreezeFrame and (2) avoidance curve by customized Matlab program.

The iphone videos were analyzed to measure the (3) total investigation time when the mouse sniffed within a 1 mm distance from the filter paper; (4) times of the risk assessment behavior (stretch-attend posture)[35]; and (5) times of flight (turned around and escape) after sniffing the filter paper. For snake skin-evoked innate fear assay, test mice were single housed in the home cage with food/water grid and bedding for overnight. A piece of molted snake skin ($2 \text{ cm} \times 16 \text{ cm}$) was introduced into a corner of the cage (do not remove the cover or the food/water grid during the test). The videos were recorded using an IPad from the side and analyzed by one experienced technician.

**Measurement of plasma stress hormone level.** The plasma level of stress hormone corticosterone was measured as previously described[7,12]. $Trpa1^{+/-}$ and $Trpa1^{-/-}$ male mice were individually housed in the home cage and habituated in the test environment overnight. The experiment was performed between 8:00 and 10:00 am. Odor exposure was performed by quietly dropping a piece of filter paper containing water or 2MT (25 µl, $2.62 \times 10^{-4}$ mole) into the home cage ($19.1 \times 29.1 \times 12.7$ cm). After odor exposure for 20 min, blood samples were collected and centrifuged at $1000 \times g$ for 15 min at 4 °C. The supernatant was collected and the plasma concentration of corticosterone was determined using an ELISA kit according to manufacturer's protocols (Assaypro, St. Charles, MO, USA).

**Open field test.** $Trpa1^{+/-}$ and $Trpa1^{-/-}$ male mice were individually placed in the periphery of a novel open field environment ($44 \times 44 \times 30$ cm) in a dimly lit room and allowed to explore for 5 min. The animals were monitored from above by a digital video camera connected to a computer running video tracking software (Ethovision 3.0, Noldus, Leesburg, VA, USA) to determine the time, distance moved and number of entries into two areas: the periphery (5 cm from the walls) and the center ($14 \text{ cm} \times 14$ cm). The open field arenas were wiped and allowed to dry between mice. Data were analyzed with the unpaired Student's $t$-test.

**Elevated-plus maze test.** $Trpa1^{+/-}$ and $Trpa1^{-/-}$ male mice were individually placed in the center of a black plexiglass elevated plus maze (each arm 30 cm long and 5 cm wide with two opposite arms closed by 25 cm high walls) elevated 31 cm in a dimly lit room and allowed to explore for 5 min. The animals were monitored from above by a video camera connected to a computer running video tracking software (Ethovision 3.0, Noldus, Leesburg, VA, USA) to determine time spent in the open and closed arms, time spent in the middle, and the number of entries into the open and closed arm. The apparatus was wiped and allowed to dry between mice. Data were analyzed with the unpaired Student's $t$-test.

**Sound-based learned fear assay.** $Trpa1^{+/-}$ and $Trpa1^{-/-}$ male mice were individually placed into a chamber equipped with a metal grid floor connected to a scrambled shock generator (Med Associates Inc., St. Albans, USA). After 1 min, the mice received a series of footshocks (2 s each) with increasing intensity. The initial shock intensity was 0.05 mA and the amplitude was increased by 0.05 mA for each consecutive footshock with 15 s intershock interval. The first shock intensity that each animal displayed each behavior (flinch, jump, and vocalization) was recorded. Once the animal displayed all three behaviors, it was removed from the chamber.

For fear conditioning, $Trpa1^{+/-}$ and $Trpa1^{-/-}$ males were individually placed in the chamber. After 2 min, the mice received three tone-shock pairings (30 s white noise, 80 dB tone co-terminated with a 2 s, 0.5 mA footshock, 1 min intertrial interval). The following day, memory of the context was assessed by placing the mice into the same chambers and freezing was measured automatically by the Med Associates software. Forty-eight hours after training, memory for the white noise cue was measured by placing the mice in a chamber with altered floor and walls, different lighting, and a vanilla smell. Freezing was measured for 3 min, then the noise cue was turned on for an additional 3 min and freezing was measured.

**Olfactory habituation–dishabituation test.** The experiment was performed as described[9] with minor modifications using 14–19-week-old $Trpa1^{+/-}$ and $Trpa1^{-/-}$ male mice. In brief, mice were habituated to a clean cage ($17.5 \times 10 \times 15$ cm) for 15 min, and then a filter paper ($2 \times 2$ cm) with 20 µl of distilled water was presented for 3 min. This presentation of distilled water was repeated four times with 1-min intervals. On the fifth trial, a filter paper scented with 2MT ($2.1 \times 10^{-11}$ or $2.1 \times 10^{-10}$ mole) was presented for 3 min. Mouse behaviors were recorded with a digital camera (30 frames/s, $640 \times 480$ pixels), and the total investigation time, defined as nasal contact within a 1 mm distance to the filter paper, was measured during 3 min exposure.

**Dual odor-based learned fear assay.** The dual odor-based fear conditioning assays were performed with 12–14-week-old male mice as previously described[7,12] with minor modifications. In brief, fear conditioning was conducted in a conditioning chamber ($24.1 \times 20.3 \times 18.4$ cm, ENV-307W-CT, Med Associates Inc., St. Albans, VT) located in a sound-attenuating box equipped with a fan, which allowed continuous removal of the odor. An odor delivery apparatus (ENV-275, Med Associates, Inc.) was modified to separately deliver two odors. $Trpa1$ mutant mice were individually placed into the conditioning chambers. After 3 min of free exploration, one of the two odors, anisole (Sigma-Aldrich, Tokyo, Japan) or

eugenol (Nacalai Tesque, Kyoto, Japan) was delivered into the conditioning chamber for 30 s. The animals were exposed to six anisole presentations randomly intermingled with six eugenol presentations and separated by 4 min inter-trial intervals in the conditioning session. The 2 s of 0.4 mA electric footshocks were delivered from the metal-rod floor and overlapped only with the last 2 s of anisole presentations, but not with eugenol presentations. $1 \times 10^{-3}$ mole eugenol and anisole were used for each animal. At the following day, *Trpa1* mutant mice were individually placed into the test cages (19.1 × 29.1 × 12.7 cm, Ancare Corporation, Cat# N10HT). After 10 min habituation (no odor), each mouse was presented sequentially with a plain filter paper (2 × 2 cm) for 10 min, a filter paper scented with eugenol ($1.25 \times 10^{-4}$ mole) for 10 min, and a filter paper scented with anisole ($1.25 \times 10^{-4}$ mole) for 10 min. Mouse behaviors were recorded and the freezing rates were quantified by FreezeFrame as described above. For 2MT/TMT-based learned fear assays, $Trpa1^{-/-}$ mice were used for fear conditioning according to the same protocol except that $1 \times 10^{-3}$ mole 2MT or $1.5 \times 10^{-4}$ mole TMT was used. Mouse behaviors were similarly analyzed except that each mouse was exposed to filter paper scented with $1.25 \times 10^{-4}$ mole eugenol, $2.7 \times 10^{-4}$ mole 2MT, or $4.0 \times 10^{-4}$ mole TMT, respectively.

**c-fos mapping by in situ hybridization (ISH)**. All procedures were followed as previously described[7]. Briefly, mice were habituated in a test cage for 2 h. A piece of filter paper scented with 15 μl water or 2MT ($1.57 \times 10^{-4}$ mole) was presented for a 30-min period. Mice were anesthetized with gaseous isoflurane and perfused with ice-cold 4% paraformaldehyde (PFA) in phosphate-buffered saline (PBS). The brains with the trigeminal nerve were dissected and fixed overnight in 4% PFA in PBS at 4 °C. The fixed brains were dehydrated in a graded ethanol and xylene series and then embedded in paraffin using an automated system (Sakura Rotary, RH-12DM; Sakura Finetek, Tokyo, Japan). Coronal sections with a thickness of 5 μm were prepared using an automatic slide preparation system (AS-200s, Kurabo, Osaka, Japan). ISH was performed using an automated system (Discovery XT, Ventana Medical Systems, Oro Valley, AZ, USA), according to the manufacturer's protocols. The digoxigenin (DIG)-labeled *Fos* probe (129-bp to 537-bp and the 543-bp to 1152-bp regions of the gene) was diluted in 1:200 and hybridized at 74 °C for 3 h using a RiboMap Kit (Roche). The slides were then incubated with biotin-conjugated anti-DIG antibody (1:500, Jackson ImmunoResearch, West Grove, PA) at 37 °C for 28 min. The probe was detected using the Ventana BlueMap Kit (Roche, Basel, Switzerland) at 37 °C for 6 h, and counterstained with a Red counterstain kit (Roche) at 37 °C for 4 min. Images were scanned using a Nano-Zoomer virtual microscope system (2.0 RS, Hamamatsu Photonics, Hamamatsu, Japan), converted to gray scale, and signal intensities were quantified using Adobe Photoshop ($n = 6$–8 for each region). Anatomical locations were determined according to the mouse brain atlas[60].

**Plasmid construction and site-directed mutagenesis**. To make the pTrpa1-P2A-mCherry construct, we amplified from mouse cDNA library the full-length Trpa1 open reading frame (ORF) by PCR with primers 5′-AAATTTCTCGAGATGAAG CGCGGCTTGAGGAG-3′ and 5′-AAATTTGGTACCAAAGTCCGGGTGGCTA ATAG-3′. The PCR product was cut with XhoI and SacII and subcloned into pEGFP-N1 to make pTrpa1. The P2A-mCherry fragment was amplified by PCR with primers 5′-AAATTTCCGCGGGGAAGCGGAGCTACTAACTT-3′ and 5′-AA ATTTGCGGCCGCTTACTTGTACAGCTCGTCCATGCCG-3′. The P2A-mCherry fragment was cut with SacII and NotI and subcloned into pTrpa1 to generate pTrpa1-P2A-mCherry.

We mutated a number of cysteines of mouse Trpa1 to serines in the pTrpa1-P2A-mCherry plasmid using PfuUltra HF DNA polymerase (Agilent, Santa Clara, CA, USA). The primers for site-directed mutagenesis were listed below:
C31S-F: 5′-GGAAGACATGGACTCCTCCAAGGAATCCTTTAAGGTGGA-3′
C31S-R: 5′-AAGGATTCCTTGGAGGAGTCCATGTCTTCCCCGAGCGCC-3′
C45S-F: 5′-TTGAAGGAGATATGTCTAGATTAGAAGACTTCATCAAGA ACC-3′
C45S-R: 5′-GAAGTCTTCTAATCTAGACATATCTCCTTCAATGTCCACCT TAAAG-3′
C66S-F: 5′-AGGATGAAAATCTCTCTCCTCTGCATCACGCAGCAGCA-3′
C66S-R: 5′-GCGTGATGCAGAGGAGAGAGATTTTCATCCTCATATTTG-3′
C89S-F: 5′-TCAATGGTTCTTCGTCTGAAGTGCTGAATATAATGGATGG-3′
C89S-R: 5′-ATATTCAGCACTTCAGACGAAGAACCATTGATGATCAGT TC-3′
C364S-F: 5′-TGAATTTGCTCCTCTCTAAAGGTGCCAAAGTAGACATAAA AGATC-3′
C364S-R: 5′-ACTTTGGCACCTTTAGAGAGGAGCAAATTCACAATGTT CC-3′
C415S-F: 5′-AAGACAATGACGGATCCACACCTCTCCATTATGCCTGT AG-3′
C415S-R: 5′-TAATGGAGAGGTGTGGATCCGTCATTGTCTTCATCCAT CA-3′
C422S-F: 5′-CCTCTCCATTATGCCTCTAGGCAGGGGGTTCCTGTCTCTG-3′
C422S-R: 5′-GGAACCCCCTGCCTAGAGGCATAATGGAGAGGTGTG-3′
C541S-F: 5′-ATACTAACTTGAAATCCACAGACCGACTAGATGAAGAA GG-3′

C541S-R: 5′-TCTAGTCGGTCTGTGGATTTCAAGTTAGTATCAAGAAT GA-3′
C622S-F: 5′-TCCAAGCAATCGATCTCCAATCATGGAGATGGTAGAATA CCTC-3′
C622S-R: 5′-TCTCCATGATTGGAGATCGATTGCTTGGAGAATTATGAG TG-3′
C633S-F: 5′-AATACCTCCCCGAGTCCATGAAAGTTCTTTTAGATTTCTG-3′
C633S-R: 5′-AAAAGAACTTTCATGGACTCGGGGAGGTATTCTACCAT CT-3′
C642S-F: 5′-TTCTTTTAGATTTCTCCATGATACCTTCCACAGAAGACAA-3′
C642S-R: 5′-GTGGAAGGTATCATGGAGAAATCTAAAAGAACTTTCAT GC-3′
C666S-F: 5′-TCAAGTATCTCCAATCCCCATTATCCATGACCAAAAAAG TAGC-3′
C666S-R: 5′-GTCATGGATAATGGGGATTGGAGATACTTGAAATTAT ACTC-3′
C714S-F: 5′-TACTCATGAAATGGTCTGCCTATGGATTCAGAGCCCATAT GA-3′
C714S-R: 5′-CTGAATCCATAGGCAGACCATTTCATGAGTAAGTACTC CCT-3′
C859S-F: 5′-AAAGGTTTGAGAACTCTGGAATTTTCATTGTTATGTTGGA GG-3′
C859S-R: 5′-ACAATGAAAATTCCAGAGTTCTCAAACCTTTGAAGATACA GTAGG-3′
C1041-F: 5′-CAAACATTGACACATCCTTGGAAATGGAAATATTGAAAC AG-3′
C1041-R: 5′-ATTTCCATTTCCAAGGATGTGTCAATGTTTGGTACTTCTTG-3′

To construct HA-tagged wild-type and mutant Trpa1 plasmids, mouse Trpa1 ORF was amplified by PCR using primers: 5′-AAATTTGGATCCAAGCGCGGC TTGAGGAGGAT-3′ and 5′-AAATTTGGTACCAAAGTCCGGGTGGCTAATA G-3′. The PCR product was digested with BamHI and KpnI, subcloned into the pKH3 plasmid, and verified by sequencing. A second PCR was performed to amplify HA-Trpa1 from pKH3-Trpa1 using primers: 5′-AAATTTCTCGAGAT GTACCCATACGATGTTCC-3′ and 5′-AAATTTCCGCGGGAGCGTAATCT GGAACATCGT-3′, cut with XhoI and SacI, and subcloned into pTrpa1-P2A-mCherry to replace the untagged Trpa1 sequence. Site-directed mutagenesis was performed on the pHA-Trpa1-P2A-mCherry construct using the same primers as listed above.

**Calcium imaging**. HEK293T cells (RCB2202) were obtained from the RIKEN BRC Cell bank and were cultured in DMEM media containing 10% FBS (GIBCO, USA) and 1% Penicillin-Streptomycin at 37 °C with 5% CO₂. We regularly tested HEK293T cells for potential mycoplasma contamination using MycoAlert$^{\text{TM}}$ mycoplasma detection kit (Lonza). For authentication, we renewed HEK293T cells regularly by purchasing new cell stocks from the RIKEN BRC Cell bank. We ensured that cells were healthy by passaged for a maximum of 3 weeks.

For calcium imaging, approximately $6.5 \times 10^5$ HEK293T cells were seeded into a 10-cm dish and passaged at 48 h before reaching confluency. UV-sterilized 96-well sensoplates with glass bottoms (Greiner Bio-One, Germany) were coated with 25 μg/ml Poly-D Lysine (Sigma-Aldrich, USA) in 40 μl calcium free Hank's Balanced Salt Solution (HBSS: 137 mM NaCl, 5.4 mM KCl, 0.44 mM KH₂PO₄, 0.34 mM Na₂HPO₄, 5.6 mM Glucose, 4.2 mM NaHCO₃, pH 7.4) for 2 h. The wells were washed twice with fresh DMEM, loaded with 100 μl DMEM containing $6.5 \times 10^3$ cells, and incubated for 18 h. For cell transfection, we mixed 5 μl Opti-MEM (Gibco, USA) with 0.2 μl Fugene 6 Transfection Reagent (Promega, USA) and 60 ng plasmid per well. For plasmids lacking mCherry marker, cells were co-transfected with 30 ng test plasmid and 30 ng mCherry plasmid per well. After 30 min incubation, 100 μl DMEM without antibiotics was added, gently mixed, and transferred to 96 wells followed by incubation for 24 h.

A 4 μM fura-2 AM (Invitrogen, USA) solution was made in modified HBSS (137 mM NaCl, 5.4 mM KCl, 0.44 mM KH₂PO₄, 0.34 mM Na₂HPO₄, 5.6 mM Glucose, 4.2 mM NaHCO₃, 20 mM HEPES, 12.4 μM Probenecid (Sigma-Aldrich, USA), 0.1% BSA, pH 7.2). After sonication for 7 min, 100 μl of 4 μM fura-2 AM solution was added to each well of the cell plate and incubated at 37 °C for 50 min. On the other hand, different 4× ligand solutions were made in calcium imaging buffer (125 mM NaCl, 2 mM MgCl₂, 4.5 mM KCl, 10 mM glucose, 20 mM HEPES, 2 mM CaCl₂, 0.1% DMSO, pH 7.2) and sonicated for 3 min. The cell plate was washed twice with calcium buffer followed by addition of 75 μl calcium buffer per well. Both the cell plate and ligand plate were incubated in the Hamamatsu calcium imaging chamber at 30 °C for 40 min to de-protect the ester form of fura-2 AM prior to calcium imaging.

Calcium imaging was performed on the Hamamatsu IMACS2 machine using a 20× lens (Olympus UPLSAPO 20× ZDC) and the IMACS imaging software (Version 1.008). After adding 25 μl of 4× ligand solution to the corresponding well in the cell plate, the fluorescence emission was continuously monitored for 5 min. The 340/380 nm fluorescence ratio (F) for each sample was normalized to that (F0) of the control sample: typically, vehicle-treated HEK293T cells that were transfected with the same plasmid(s). The IMACS2 machine automatically recorded the 340/380 nm ratio of mCherry-expressing cells. During post-imaging analysis, the mCherry-positive cells with abnormal or unhealthy morphology were

excluded. The average fluorescent ratio of all mCherry-positive cells was measured at every 2 s interval. The quantitative results of calcium imaging for wild-type and mutant Trpa1 proteins were graphed with Prism 7.0 (GraphPad, CA, USA). For confirmation of the expression and functionality of various Trp channels, control compounds were used to stimulate $Ca^{2+}$ responses as indicated[40]. For inhibition of Trpa1 with antagonist, transfected HEK293T cells were pre-incubated for 10 min in 50 μM solutions of HC-030031 (Sigma-Aldrich) dissolved in calcium imaging buffer with 0.2% DMSO, which was vortexed immediately before adding 2MT.

**Synthesis of TMT, alkynyl-TMT, and SBT**. All distillations were carried out with a Büchi Glass Oven B-585 Kugelrohr. $^1H$ and $^{13}C$ NMR spectra data were obtained with JEOL JNM-ECS 400 instruments. Chemical shifts ($\delta$) are quoted in parts per million using tetramethylsilane ($\delta = 0$ ppm) as the reference for $^1H$ NMR spectroscopy, and $CDCl_3$ ($\delta = 77.0$ ppm) for $^{13}C$ NMR spectroscopy. Reagents and solvents were purchased from the following commercial suppliers: Tokyo Chemical Ind., Sigma-Aldrich, Inc., Kanto Chemical Co., Inc., Wako Pure Chemical Ind., Ltd., and Nacalai Tesque. Thin-layer chromatography was performed on Merck Kieselgel 60 F254 (0.25 mm) plates. Synthetic schemes of TMT (**3a**)[61] and alkynyl-TMT (**3b**) are shown in Supplementary Fig. 8a and the methods are described here:

*Synthesis of 2,4,5-trimethyl-2,5-dihydrothiazole (TMT, **3a**)*: To a stirring mixture of 3-mercapto-2-butanone (**1**, 2.1 ml, 21.1 mmol) and acetoaldehyde (**2a**, 1.18 ml, 21.1 mmol) was added $NH_3$ gas at 0 °C over a period of 2 h (until the reaction which releases water and heat is finished). After being diluted with diethyl ether, the inorganic phase was separated. The organic phase was washed with brine, dried over $Na_2SO_4$, and concentrated at pressures above 400 MPa. The residual oil was distilled with Kugelrohr distillation apparatus (1 mmHg, 50–70 °C) to afford **3a** as a 43:57 mixture of conceivable isomers (1.2 g, 9.3 mmol, 44%, colorless oil).

*Minor isomer of **3a***. $^1H$ NMR (400 MHz, $CDCl_3$) $\delta$ 5.61–5.52 (m, $CH_3$-CH(C)-S, 1H), 4.37–4.20 (m, N-CH($CH_3$)-S, 1H), 2.06 (m, $CH_3$-C(CH) = N, 3H), 1.56 (dd, $J = 6.6$, 1.9 Hz, $CH_3$, 3H), 1.46 (dd, $J = 7.1$, 1.8 Hz, $CH_3$, 3H). $^{13}C$ NMR (101 MHz, $CDCl_3$) $\delta$ 172.22, 75.53, 56.55, 25.24, 21.72, 17.71.

*Major isomer of **3a***. $^1H$ NMR (400 MHz, $CDCl_3$) $\delta$ 5.52–5.44 (m, $CH_3$-CH(C)-S, 1H), 4.37–4.20 (m, N-CH($CH_3$)-S, 1H), 2.06 (m, $CH_3$-C(CH) = N, 3H), 1.58 (dd, $J = 6.6$, 1.9 Hz, $CH_3$, 3H), 1.51 (dd, $J = 7.1$, 1.8 Hz, $CH_3$, 3H). $^{13}C$ NMR (101 MHz, $CDCl_3$) $\delta$ 172.01, 75.82, 56.78, 26.24, 22.62, 17.68.

*Synthesis of 2-(but-3-yn-1-yl)-4,5-dimethyl-2,5-dihydrothiazole (Alkynyl-TMT, **3b**)*: To a solution of 4-pentyn-1-ol (1 ml, 10.8 mmol) in $CH_2Cl_2$ (15 ml), trimethylamine (10 ml), DMSO (6 ml), and $SO_3$ pyridine complex (5.2 g, 32.4 mmol) were added at 0 °C, and the mixture was stirred at same temperature for 3 h. After being neutralized by 1.0 M aq. hydrochloric acid, the mixture was extracted with chloroform, washed with brine, dried over $Na_2SO_4$, and concentrated under reduced pressure. The residual oil was distilled with Kugelrohr distillation apparatus (760 mmHg, 100–150 °C) to afford 4-pentyn-1-ol (**2b**, 400 mg, 4.9 mmol, 45%, colorless oil).

To a stirring mixture of 3-mercapto-2-butanone (**1**, 474 μl, 4.9 mmol) and **2b** (400 mg, 4.9 mmol) was added $NH_3$ gas at 0 °C over a period of 2 h (until the reaction which releases water and heat is finished). After being dissolved the white precipitation with ethyl acetate, the inorganic phase was separated. The organic phase was washed with brine, dried over $Na_2SO_4$, and concentrated at reduced pressure. The residual oil was distilled with Kugelrohr distillation apparatus (1 mmHg, 50–90 °C) to afford the titled compound as a 3:7 mixture of conceivable isomers (180 mg, 1.1 mmol, 22%).

*Minor isomer of **3b***. $^1H$ NMR (400 MHz, $CDCl_3$) $\delta$ 5.67–5.62 (m, $CH_3$-CH(C)-S, 1H), 4.28–4.21 (m, N-CH(Alkyl)-S, 1H), 2.38–2.33 (m, $-CH_2-$, 2H), 2.18–2.09 (m, $-CH_2-$, 1H), 2.06–2.05 (m, $CH_3$-C(CH)=N, 3H), 1.97–1.95 (m, C≡C–H, 1H), 1.96–1.86 (m, $-CH_2-$, 1H), 1.46 (d, $J = 7.1$ Hz, $CH_3$, 3H). $^{13}C$ NMR (101 MHz, $CDCl_3$) $\delta$ 172.92, 83.48, 79.44, 68.93, 55.85, 37.43, 21.64, 17.68, 15.76.

*Major isomer of **3b***. $^1H$ NMR (400 MHz, $CDCl_3$) $\delta$ 5.59–5.54 (m, $CH_3$-CH(C)-S, 1H), 4.28–4.21 (m, N-CH(Alkyl)-S, 1H), 2.38–2.33 (m, $-CH_2-$, 2H), 2.18–2.09 (m, $-CH_2-$, 1H), 2.06–2.05 (m, $CH_3$-C(CH)=N, 3H), 1.97–1.95 (m, C≡C–H, 1H), 1.96–1.86 (m, $-CH_2-$, 1H), 1.51 (d, $J = 7.0$ Hz, $CH_3$, 3H). $^{13}C$ NMR (101 MHz, $CDCl_3$) $\delta$ 172.74, 83.48, 79.79, 68.98, 56.05, 38.12, 22.41, 17.68, 15.86.

SBT was synthesized as previously reported[62]. And the method is described here. To a suspension of 2-aminoethanethiol hydrochloride (**4**, 3.4 g, 30.0 mmol) in dry toluene (125 ml) under argon, a solution of triisobuthylaluminium (1.0 M in toluene, 75.0 ml, 75.0 mmol) was added dropwise at room temperature. After refluxing for 30 min, methyl DL-2-methylbutyrate (**5**, 3.3 g, 28.5 mmol) was added dropwise. After further refluxing for 2 h, the mixture was diluted with 125 ml of toluene, cooled to room temperature, and quenched with MeOH (50 ml). The mixture was stirred at room temperature for 30 min, and then a sat. aq. Rochelle's salt (100 ml), sat. aq. $NaHCO_3$ (100 ml), and brine (100 ml) were successively added and stirred for 1 h. After being diluted with diethyl ether, the inorganic phase was separated. The organic phase was washed with brine, dried over $Na_2SO_4$, and concentrated in vacuo. The residual oil was distilled with Kugelrohr distillation apparatus (1 mmHg, 60–120 °C) to afford **6** (1.82 g, 12.74 mmol, 42%, colorless oil).

$^1H$ NMR (400 MHz, $CDCl_3$) $\delta$ 4.21 (ddt, $J = 8.3$, 0.8 Hz, 2H), 3.24 (t, $J = 8.3$ Hz, 2H), 2.68–2.58 (m, 1H), 1.74–1.45 (m, 2H), 1.19 (d, $J = 6.8$ Hz, 3H), 0.92 (t, $J = 7.3$ Hz, 3H). $^{13}C$ NMR (101 MHz, $CDCl_3$) $\delta$ 176.92, 64.38, 40.93, 33.11, 28.64, 18.84, 11.78.

**Streptavidin pulldown of alkynyl-TMT-modified HA-Trpa1**. Approximately 1 × $10^5$ HEK293T cells were seeded into a 12-well plate and incubated in DMEM medium containing 10% FBS and 1% Penicillin/Streptomycin at 37 °C for 18 h. For transient transfection, 1 μg of pHA-Trpa1-P2A-mCherry construct was mixed with 3 μl Fugene 6 in 60 μl OptiMEM per 12 well. After 24 h of transfection, the cells were washed with 500 μl calcium solution (125 mM NaCl, 4.5 mM KCl, 10 mM glucose, 20 mM HEPES, 2 mM $CaCl_2$, 0.1% DMSO, pH 7.2), and incubated with 300 μM alkynyl-TMT in calcium solution at 30 °C for 6 min. After washing again with calcium solution, the cells were lysed with lysis buffer (50 mM HEPES, 2 mM $MgCl_2$, 10 mM KCl, 1% SDS, 150 units/ml of Benzonase (Sigma-Aldrich, USA) pH 8.0, supplemented with complete™, Mini, EDTA-free protease inhibitor cocktail (Sigma-Aldrich, USA)) for 20 min at room temperature. The lysate was centrifuged at 12,000 × $g$ for 3 min to pellet cell debris and the protein concentration was measured by BCA assay (Thermo Fisher Scientific, MA, USA).

For click chemistry reaction, 2 μl of 50 mM $CuSO_4$ solution was mixed with 2 μl of 2.5 mM tris(3-hydroxypropyltriazolylmethyl)amine (THPTA, Sigma-Aldrich, USA), followed by addition of 1 μl of 1.25 mM azide-PEG3-biotin conjugate (Sigma-Aldrich, USA) and 1 μl of 50 mM tris(2-carboxyethyl)phosphine (TCEP, Sigma-Aldrich, USA). The 4 μl mixture of $CuSO_4$/THPTA complex was then mixed with 44 μl cell lysates (equal amount of total proteins) by vortexing and incubated at room temperature for 2 h with constant agitation. Afterwards, equal amount of protein samples (~20–40 μg) were diluted with RIPA buffer (10 mM Tris HCl, 140 mM NaCl, 1 mM EDTA, 0.5 mM EGTA, 0.1% SDS, 0.5% sodium deoxycholate, 1% Triton X-100, pH 8.0) containing protease inhibitor cocktail and incubated with 15 μl of streptavidin agarose (Cat# 69203, EMD Millipore, Germany) for 2 h at room temperature with constant rotating. After washing the streptavidin beads three times with 150 μl RIPA buffer, the biotin-conjugated proteins were eluted with 1× SDS loading buffer with 50 mM DTT at 95 °C for 10 min. The eluted proteins were resolved by SDS-polyacrylamide gel electrophoresis, transferred to nitrocellulose membrane, and immunoblotted with rabbit anti-HA antibody (1:50,000, Cat# 3724s, Cell Signaling Technology, USA) or mouse anti-β-actin antibody (1:50,000, Cat# sc-47778, Santa Cruz Biotechnology, USA).

**c-Fos IHC and double *Trpa1*/c-Fos staining of TG**. Mice were single housed and habituated in the test environment 24 h before the experiment. A small filter paper containing 15 μl of water or 2MT ($1.57 × 10^{-4}$ mole) was placed into the home cage for 30 min. After the filter paper was removed, test mouse was allowed to recover in the home cage for another 30 min. Mice were then rapidly anesthetized with pentobarbital (50 mg/kg, i.p.) and sequentially perfused by PBS and 4% PFA in PBS. Trigeminal ganglions were dissected and fixed for 24 h in the same fixative solution and stored at 4 °C, and then cryoprotected with 30% sucrose (wt/vol) in PBS for 48 h. The tissues were put in the Tissue-Tek O.C.T compound (Sakura Finetek), and 30-μm-thick coronal sections were cut on a cryostat (CM3050S, Leica). For c-Fos IHC, the floating TG sections were washed three times with PBS for 5 min, incubated with 0.3% Triton X-100 in 0.01 M PBS for 15 min, and washed again with PBS for three times. The slides were treated with 1% $H_2O_2$/PBS for 30 min and washed three times with PBS. After incubating in 0.3% of BSA/0.25% Triton-X-100/ PBS for 1 h, the sections were incubated with rabbit anti-c-Fos antibody (1:5000, EMD Millipore, ABE457) at 4 °C overnight. After washing six times with PBST, the sections were incubated with biotinylated rabbit anti-goat IgG (1:500, Vector BA5000) for 2 h at room temperature and followed by PBS washing for three times. The sections were incubated with HRP-Streptavidin (ABC reagent, VECTOR) for 1 h at room temperature. After washing with PBS three times, the slides were stained in DAB Peroxidase (HRP) Substrate Kit (VECTOR) in the dark until the color developed. The reaction was stopped by PBS washing and the sections were mounted onto glass slides for microscopy. All images were acquired using the Zeiss LSM700 confocal microscope with 10× objective lens (NA = 0.45) under the AxioVision4.8 software. The nuclear c-Fos + neurons were counted in all sections from the same TG. Representative images shown in the figures were chosen from a similar region of the trigeminal ganglion based on morphology.

For double *Trpa1*/c-Fos staining of TG, *Trpa1* mRNA ISH was performed before c-Fos IHC. All reagents and solutions were prepared using diethylpyrocarbonate (DEPC)-treated distilled deionized water (ddH$_2$O) and RNase-free reagents. The DNA templates for making sense or antisense *Trpa1* probes were generated using PCR primers: 5′-GCCTAATACGACTCACTATA GGCGGCTTGAGGAGGATTCTG-3′ and 5′-TCTGTGAAGCAGGGTCTCCT-3′; or 5′-GCGGCTTGAGGAGGATTCTG-3′ and 5′-GCCTAAGACTCACTATAGG GTCTGTGAAGCAGGGTCTCCT-3′. The RNA probes labeled with digoxygenin-UTP were generated by in vitro transcription by T7-RNA labeling kit (Roche). The floating TG sections were washed with 1× PBS for 10 min and treated with PFA (4% in PBS, wt/vol, Nacalai) for 15 min, then the slides were reduced by treatment of 0.3% of sodium borohydride and washed with PBS 10 min. Tissue slides were treated with 0.3% Triton X-100 (vol/vol) in PBS for 15 min followed by twice of PBS washing. The slides were then treated with 0.75% glycine for 15 min followed by PBS washing and acetylated by adding 1/400 vol of acetic anhydride (vol/vol) in PBS for 15 min, followed by PBS washing two times. Afterward, the tissue sections were incubated in the hybridization solution (50% deionized formamide, 2% blocking reagent, 5× SSC, 0.1% N-lauroylsarcosine, 0.1% SDS) without probes for 1 h at 55 °C, followed by incubation in the hybridization solution with the *Trpa1*

antisense probe for 16–18 h at 60 °C. After hybridization, slides were washed with 2× SSC/50% formamide/0.1% N-lauroylsarcosine for 20 min at 50 °C followed by treatment with 5 μg/ml RNaseA in PBS at 37 °C for 30 min. The slides were then washed with 2× SSC/0.1% N-lauroylsarcosine for 30 min at 37 °C, followed by washing twice with 0.2% SSC/0.1% N-lauroylsarcosine for 30 min at 37 °C. The slides were rinsed with TS 7.5 for 5 min and incubated with 1% blocking reagent (Roche) diluted in TS7.5 for 1 h. After blocking, the slides were incubated with 1:2000 Anti-DIG-AP antibody (Roche, 11093274910) for 16–18 h at 4 °C. Next day, the sections were washed three times in TS7.5/0.1% Tween 20 for 20 min each and were incubated in TS9.5 for 5 min. Finally, the slides were incubated in 1:50 diluted NBT/BCIP solution (Roche) in TS9.5 in the dark until the color was developed. After staining was stopped by PBS washing, c-Fos IHC staining was performed as described above.

**Unilateral lesion of the trigeminal ganglion**. Mice were anesthetized with pentobarbital (50 mg/kg, i.p.) and mounted on a stereotaxic apparatus (Narishige Group, Japan). The skin was opened, the skull was exposed, and three holes were made over the right hemisphere using a dental drill. An electrode was lowered into the trigeminal ganglion (coordinates: A-P, +1.5 mm; M-L, +0.8 mm; D-V, −6.5 mm; A-P, +0.9 mm; M-L, +1.0 mm; D-V, −6.5 mm; and A-P, +0.9 mm; M-L, +1.2 mm; D-V, −6.5 mm from bregma) in both the sham and the unilateral lesion (ulTGx) groups. In the ulTGx group, a constant current (2 mA) was applied to the trigeminal ganglion for 15 s. The electrode was removed and the skin was closed. After recovery for 8 days, 2MT-evoked freezing behavior was measured for the sham and ulTGx mice (n = 8 each) as described above. After testing, mice were presented with filter papers spotted with 2MT, sacrificed, and c-fos mRNA expression was analyzed. Lesioned areas were verified by the histological examination of coronal sections.

**Stereotaxic viral injections and behavioral test**. Mice were deeply anesthetized with 2% isoflurane and positioned on stereotaxic apparatus (Kopf Instruments, Tujunga, CA). Small holes were drilled into the skull and injections of AAV-GFP or AAV-Trpa1 were performed bilaterally using a Hamilton 0.5 μL Neuros Model 7000.5KH syringe with 33GA needles (Hamilton, USA) with 800 nl of virus solution (titer: ~1.0 × 10^13 viral genomes (VG)/ml for AAV-Trpa1 and ~1.0 × 10^14 viral genomes (VG)/ml for AAV-GFP) at a flow rate of 70 nl/min. Coordinates for injection are: A-P, +0.9 mm; M-L, ±1.4 mm; D-V, −6.5 mm from bregma. Both AAV-GFP and AAV-Trpa1-injected $Trpa1^{-/-}$ mice were subjected to 2MT-evoked innate fear assay 2 months after virus administration.

**Statistical methods**. Values are expressed as mean ± SEM. Data were tested for Gaussian distribution and variance and it fits the normal distribution pattern. Between different groups, we tested their similarity with Levene's test. For Figs. 3c and 4d, we performed two-way ANOVA followed by Tukey multiple comparisons test. For Fig. 3a, b and Supplementary Fig. 5a, we performed one-way ANOVA followed by Tukey multiple comparisons test as geometric mean ± 95% confidence limits in Graphpad Prism (La Jolla CA, USA). Statistical significance was calculated using the unpaired two-tailed Student's $t$-test in excel for Figs. 6b and 7c, e, k. The two-tailed unpaired Student's $t$-test was calculated with 95% confidence interval with Graphpad Prism for the following figures: 3e, g, 4b, c, 6d, Supplementary Fig. 4a, Supplementary Fig. 4b, Supplementary Fig. 4c, Supplementary Fig. 4d, Supplementary Fig. 5d, Supplementary Fig. 6. For Fig. 5d and e, pairwise comparisons were made for each mutant Trpa1 to wild-type Trpa1 control using two-way unpaired Student's $t$-test. Multiple comparisons were controlled by Bonferroni correction. For Supplementary Fig. 7g, the response of each mutant plasmid was compared to a wild-type control tested for 2MT or TMT for individual Student's $t$-tests in excel. For Figs. 4e and 7d, j, significance was calculated by Student's $t$-test in excel comparing the freezing rate for the experimental group to the control group for that time unit. For Fig. 2g, the three different comparisons were calculated between each pair of groups using the Student's $t$-test in excel. Significance levels are indicated as follows: $*P < 0.05$; $**P < 0.01$; $***P < 0.001$; ns not significant ($P > 0.05$).

**Data availability**. We claim all relevant data are available from the authors with reasonable request.

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

## Acknowledgements

We thank Drs. Y. Mori, R. Sakaguchi, M. Ding, T. Han, and J. Chen for sharing plasmids, Dr. Bart Carter for providing snake skin molts, and Drs. J. Cohen, H. Hu, and Y. Liu for comments on our manuscript. We acknowledge R. Potts, Y. Ogawa, W. Wang, T. Liang, H. Kim, and S. Kim for technical assistance. Q.L. is a W.A. "Tex" Moncrief Jr. Scholar in Biomedical Research. This work was supported by the National Institute of Health (GM111367 to Q.L.; R37GM067759 to B.B.), the Welch Foundation (I-1608 to Q.L.) and UT Southwestern Medical Center (High Impact/High Risk grant to Q.L.) as well as the Japan Society for the Promotion of Science (15K14874 and "Willdynamics" 17H06048 to Q.L.; 26220207 to M.Y., H.M.; 16H02591 to K.K.; 17K08133 to L.C.), the Japan Science and Technology Agency-step grant (AS271516U to K.K., R.K.), and the World Premier International Research Center Initiative (WPI) from MEXT, Japan.

## Author contributions

Q.L., B.B., K.K. designed the experiments. L.C. optimized 2MT-evoked innate fear assay for genetic screening. B.B., J.R., T.W., X.Z. designed the screening platform and generated ENU mutant mice. K.W. and Y.W. conducted large-scale innate fear screening. Y.W., L. C., C.L., T.M., K.W., L.T., G.A., D.K-N., L.W., J.Z., Y.L., C.M. performed genetics, biochemical, and behavior experiments with input from T.S., K.S., Y.I., M.Y., H.F., J.C., J.P. M., L.G., T.M., E.H., Y.C., Z.W., and R.K. performed whole brain *c-fos* mapping. C.L., S. S., K.S. performed *Trpa1*/c-Fos TG staining and AAV rescue experiments. T.S. and H.N. synthesized TMT, alkynyl-TMT, and SBT. Y.W. prepared the figures and Q.L. wrote the manuscript.

## Additional information

**Competing interests:** The authors declare no competing interests.

