## [Peer Review File · Nature Communications]

Editorial Note: Reviewer #3 was added to the reviewer panel during the second round of peer review with expertise in large-scale forward genetics screening in mice.

Reviewers' comments:

Reviewer #1 (Remarks to the Author):

In this study Wang and colleagues found, through a forward genetics screen, that TRPA1 receptors are crucial for innate defensive responses to TMT (a component of fox feces) and 2MT (a TMT analogue). Null mutations in this gene impaired freezing in response to TMT or 2MT. This result was confirmed using a knock out mouse for the *trpa1* gene. In addition, the authors show that activation of areas previously shown to respond to TMT, such as central amygdala and ventro-lateral PAG, is TRAP1-dependent. Using the heterologous expression system the authors found calcium transients in HEK cells expressing the TRAP1 receptor, suggesting that these receptors may function as chemosensors, a function that depends on a few cysteine residues. The authors further show that TRAP1-dependent responses to TMT/2MT are mediated by trigeminal neurons. Unilateral ablation of the trigeminal nucleus decreases innate freezing to 2MT and rescue of the *trpa1* KO in trigeminal neurons rescues the behavioral deficits.

This is an impressive work that brings novel insights into the mechanisms of innate defensive responses to odors. I have however a few concerns the authors should address:

- 1) The authors convincingly show that innate freezing triggered by TMT/2MT depends on TRPA1 receptors. However, there is a debate around the validity of TMT as a predator odor. Although it is interesting that innate freezing to this odorant depends on TRPA1 receptors, it would be great to know whether freezing triggered by natural predator odors taps onto the same mechanism. To address this issue the authors could perform a simple experiment testing *trpa1* mutant mice to natural predator odors, such as fox feces (from which TMT was isolated) or cat odor. Even if freezing is not reliably triggered by these odors in WT mice, they can look at avoidance. Disruption of innate defensive responses to natural predator odors in *trpa1* KO mice would strengthen their argument that *trpa1* mediates chemical detection of predators. If TRPA1 receptors are not required for defensive responses, freezing or avoidance, to predator odors, it is still interesting.
- 2) The authors demonstrate that *trpa1* mutants still detect 2-MT, showing *c-fos* expression in the olfactory bulb and intact conditioned freezing to 2MT after pairings with footshock. However, it is unclear which part of the bulb is activated by 2MT. Given that previous work from the Sakano lab (including one of the co-authors of this study) has shown that the dorsal portion of the olfactory bulb is involved in innate responses to TMT, it would be interesting to know whether KO mice for TRAP1 show normal responses to 2MT. Could TRPA1 receptors be important for responses in the dorsal portion of the bulb? The authors have the data. Answering this question only requires analyzing *c-fos* expression in the dorsal portion of the olfactory bulb.
- 3) The authors should report their statistical analysis more clearly and include a section in the methods about it. This is particularly relevant given that in a few cases it is unclear whether the statistical tests are appropriate for the data. For example for the data in figure 4c, the authors report a one-way ANOVA for an *n* of 3. Not only the sample size is too small for an ANOVA, but it is unclear what is exactly the data the authors are trying to explain since they show 2 data sets for each receptor (one for TMT and another for 2MT). Is the data being pooled? How? When multiple comparisons are shown (* symbols in figures) but a single value of an ANOVA is reported it is unclear which post-hoc tests were used and if in all cases correction for multiple comparisons was used.
- 4) The authors only tested the rescue of *trpa1* in trigeminal neurons for responses to 2MT, the most pungent stimulus. I believe that ideally the authors should have tested rescue to TMT or 2MT low dose. Given that repeating that experiment is not trivial the authors should mention the possibility

that *trpa1* in other tissues would mediate the response to TMT or low dose 2MT, such that the rescue only in trigeminal neurons would not work. The authors mention a possible role of *trpa1* in other tissues, but not directly regarding the rescue experiment. This is relevant in light of the general discussion in the field regarding predator versus pungent odors driving innate freezing.

5) The double staining images shown for the rescue experiment (Fig7 G and H) are not very helpful without a quantification of double-labeled cells.

Reviewer #2 (Remarks to the Author):

This interesting study proposes a novel unconventional model for the signaling pathways controlling fear-like responses in mice to predator odors that involves the trigeminal ganglia and TRPA1 ion channel. The study attacks a long-standing controversy about the real role of TMT in mice: a fear-inducing chemosignal or just as an aversive pungent chemical. Three major questions arise from the results: 1) is TRPA1 a sensor for TMT/2MT? 2) are *Trpa1*^{-/-} mice still able to detect TMT/2MT? 3) are TRPA1+ cells in the TG driving fear-like responses?

1. The first question seems partially resolved, at least in HEK cells. However, increase of *c-fos* expression after 2MT exposure was detected in only a fraction of TRPA1+ neurons on the TG and many of the activated TG neurons lacked *Trpa1* expression (fig 6f), rising concerns about sensitivity and specificity. More evidence showing that *Trpa1* neurons indeed detect TMT/2MT and their level of specificity would strengthen the conclusions. Using a more physiological model or *in vivo* recordings would address this point, perhaps including other *Trpa1*-expressing neurons such as DRG. It would also help substantially to use an experimental approach with a better temporal resolution than *c-fos* staining (Ca-imaging or electrophysiology recordings, for example). These experiments should clarify the specificity of TMT/2MT for *Trpa1*-neurons comparing to potential responses in KO controls.

2. Results using the learned fear assay (Fig 5e and suppl fig 8c,d) indicate that *Trpa1*^{-/-} mice are indeed able to detect 2MT, likely via the olfactory system (Fig 5c,d). Unfortunately, no data on heterozygous or WT controls in the learned fear assay was included for comparison with the KO. This is an important control to exclude any possible reduction in olfactory performance in *Trpa1*^{-/-} mice. Reduced olfaction may have an important impact on the behavior and thus, conclusions would need to be reformulated. The cookie test shown on suppl fig 8a is clearly insufficient to evaluate olfaction in *Trpa1*^{-/-}. A habituation-deshabituation test using low 2MT dose (as on Fig 3) might be more appropriate in this case. In this context, showing that other odorants not detected by *Trpa1* (such as those used in refs 17, 47, 48) can induce freezing would greatly strengthen the work.

3. Whether *Trpa1*+ cells in the TG are driving the observed fear-like responses is unclear to me. First, the 2MT effect on behavior seems to be partial: background (no odor) freezing rates in WT mice ranges 0-20% (Fig 1a) whereas in 2MT-induced in *Trpa1* mutants is 2-fold higher, up to 20-40% (figs 2b and 3a). This would be consistent with decreased olfaction. A no odor control in *Trpa1*^{-/-} mice may clarify this point. Second, authors use a (>2s) decrease in locomotor activity as a proxy for freezing, which is assumed as a fear response. Careful observation of the movies reveals that TMT/2MT not only elicits freezing, but also sitting in distant areas from the stimulus, which can be also interpreted as avoidance. This point should be mentioned in the discussion, which now is overly oriented towards emotional fear. The "emotional" component of aversion is supposed to be lower than fear, fitting better with the classical function of TG neurons. Third, and related with the previous point, it has been reported (in ref. 13) that a subset of olfactory-activated neurons from the cortical amygdala plays a critical role in the generation of innate TMT-driven aversive behavior. Activity in this brain region in

Trpa1^{-/-} mice in response to TMT/2MT should be reported to determine whether olfaction is dispensable for the display of the innate aversive-freezing behaviors.

Minor points:

1. In fig 4 and suppl fig 6, can responses to TMT/2MT be eliminated using a specific inhibitor for Trpa1 such as HC-030031?
2. Freezing rate results on Trpv1^{-/-} mice in response to 2MT differ considerably between fig 3b (80% approx.) and suppl fig 4f (50%).

Reviewer #1 (Remarks to the Author):

1) The authors convincingly show that innate freezing triggered by TMT/2MT depends on TRPA1 receptors. However, there is a debate around the validity of TMT as a predator odor. Although it is interesting that innate freezing to this odorant depends on TRPA1 receptors, it would be great to know whether freezing triggered by natural predator odors taps onto the same mechanism. To address this issue the authors could perform a simple experiment testing *trpa1* mutant mice to natural predator odors, such as fox feces (from which TMT was isolated) or cat odor. Even if freezing is not reliably triggered by these odors in WT mice, they can look at avoidance. Disruption of innate defensive responses to natural predator odors in *trpa1* KO mice would strengthen their argument that *trpa1* mediates chemical detection of predators. If TRPA1 receptors are not required for defensive responses, freezing or avoidance, to predator odors, it is still interesting.

RE: We thank the reviewer for this excellent suggestion. We successfully showed that *Trpa1*^{-/-} mice were defective for snake skin-evoked innate freezing, avoidance, flight, and risk assessment behaviors as compared to *Trpa1*^{+/-} mice (**Fig. 3f,g**). This critical result establishes that TRPA1 mediates natural predator odor-evoked innate fear/defensive responses.

2) The authors demonstrate that *trpa1* mutants still detect 2-MT, showing *c-fos* expression in the olfactory bulb and intact conditioned freezing to 2MT after pairings with footshock. However, it is unclear which part of the bulb is activated by 2MT. Given that previous work from the Sakano lab (including one of the co-authors of this study) has shown that the dorsal portion of the olfactory bulb is involved in innate responses to TMT, it would be interesting to know whether KO mice for TRPA1 show normal responses to 2MT. Could TRPA1 receptors be important for responses in the dorsal portion of the bulb? The authors have the data. Answering this question only requires analyzing *c-fos* expression in the dorsal portion of the olfactory bulb.

RE: We previously quantified *c-fos* expression in the posterior region of the olfactory bulb (OB) (**Fig. 4a,b and Supplementary Fig. 6b**). As suggested by the reviewer, we also examined the dorsal region of OB that was specifically activated by TMT. 2MT exposure induced a similar low level of *c-fos* expression in the dorsal region of OB in *Trpa1*^{+/-} and *Trpa1*^{-/-} mice (**Supplementary Fig. 6c**).

3) The authors should report their statistical analysis more clearly and include a section in the methods about it. This is particularly relevant given that in a few cases it is unclear whether the statistical tests are appropriate for the data. For example for the data in figure 4c, the authors report a one-way ANOVA for an n of 3. Not only the sample size is too small for an ANOVA, but it is unclear what is exactly the data the authors are trying to explain since they show 2 data sets for each receptor (one for TMT and another for 2MT). Is the data being pool? How? When multiple comparisons are shown (* symbols in figures) but a single value of an ANOVA is reported it is unclear which post-hoc tests were used and if in all cases correction for multiple comparisons was used.

RE: As suggested by the reviewer, we include a section in the method about statistical analysis. We are sorry for the confusion in the original **Fig. 4c**. In the revision, we separated this figure into 2 figures for 2MT and TMT, respectively (**Fig. 5d, e**). There were at least 3 biological replicates for each *Trpa1* construct for 2MT or TMT experiments, which was normalized to wild-type construct.

4) The authors only tested the rescue of *trpa1* in trigeminal neurons for responses to 2MT, the most pungent stimulus. I believe that ideally the authors should have tested rescue to TMT or 2MT low dose. Given that repeating that experiment is not trivial the authors should mention the possibility that *trpa1* in other tissues would mediate the response to TMT or low dose 2MT, such that the rescue only in trigeminal neurons would not work. The authors mention a possible role of *trpa1* in other tissues, but not directly regarding the rescue experiment. This is relevant in light of the general discussion in the field regarding predator versus pungent odors driving innate freezing.

RE: We thank the reviewer for this excellent suggestion. Yes, we did test this, but AAV-GFP and AAV-TRPA1 injected mice showed no apparent difference in the freezing response to low dose 2MT exposure. Thus, this was probably a partial rescue because TRPA1 might have important functions in other tissues. Future studies are needed to further elucidate the specific roles of TRPA1 in the TG and other tissues with regard to predator odor-evoked innate fear/defensive responses.

5) The double staining images shown for the rescue experiment (Fig7 G and H) are not very helpful without a quantification of double-labeled cells.

RE: The reviewer has an excellent point. In this AAV rescue experiment, my students encountered difficulty in performing the perfusion and double *Trpa1* ISH/c-Fos IHC staining. They wasted most of the precious TG samples and ended up only with one good *Trpa1*/c-Fos double staining result. We now include this data in the revision (**Fig. 7i**), which shows high percentage of *Trpa1*/Fos double positive cells in AAV-TRPA1 injected TG, but not AAV-GFP injected TG. Because our remaining *Trpa1*^{-/-} mice are old, it will take us 4-6 months to generate new *Trpa1*^{-/-} mice and repeat this double staining experiment. We sincerely appreciate reviewer for his/her understanding.

Reviewer #2 (Remarks to the Author):

1. The first question seems partially resolved, at least in HEK cells. However, increase of c-fos expression after 2MT exposure was detected in only a fraction of TRPA1+ neurons on the TG and many of the activated TG neurons lacked *Trpa1* expression (fig 6f), rising concerns about sensitivity and specificity. More evidence showing that *Trpa1* neurons indeed detect TMT/2MT and their level of specificity would strengthen the conclusions. Using a more physiological model or *in vivo* recordings would address this point, perhaps including other *Trpa1*-expressing neurons such as DRG. It would also help substantially to use an experimental approach with a better temporal resolution than c-fos staining (Ca-imaging or electrophysiology recordings, for example). These experiments should clarify the specificity of TMT/2MT for *Trpa1*-neurons comparing to potential responses in KO controls.

RE: Trigeminal ganglia (TG) contain neuron cell bodies for the 3 branches (ophthalmic, mandibular, and maxillary divisions) of the trigeminal nerve. TG neurons innervate the majority of craniofacial region, including the nose, mouth, and eyes. We believe that only a subset of TG neurons that innervated specific craniofacial region, such as the nasal cavity, would be responsible for 2MT sensing. Accordingly, we observed that only a specific subset of TRPA1+ neurons showed nuclear c-Fos signals after 2MT exposure, and that only a subset of ectopic TRPA1-expressing neurons were c-Fos+ in response to 2MT in our AAV-TRPA1 TG rescued *Trpa1*^{-/-} mice (**Fig. 7h,i**). Moreover, the Fos+/TRPA1- TG neurons may be the result of intraganglion neural transmission as previously reported (Pagadala et al., J Neurosci 2013, Kung et al., Plos One 2013). Previously, two groups observed intraganglion transfer of WGA or pseudotyped-rabies virus between dorsal root ganglia (DRG) neurons (Zhang et al., J clin Invest 2015, Braz et al., J Comp Neurol 2013). Other group observed intraganglion release of neuromodulators, such as CGRP and ATP (Zhang et al., PNAS 2007, Ceruti et al., J Neurosci 2011). Notably, activation of TRPA1 regulates the secretion of neuropeptide CGRP (Eberhardt et al., Nat. commun. 2015; Pozsgai et al., Eur J Pharmacol. 2012; Quell et al., pharmacol Res Perspect. 2015). These observations, together with our findings, suggest that 2MT could activate a subset of TRPA1+ primary sensory neurons, which then release neuromodulators to stimulate c-fos expression in downstream TRPA1- neurons in the TG. Because TG is located at the base of the brain, it is highly challenging to perform *in vivo* Ca-imaging or electrophysiology

recordings. We think these are excellent experiments to pursue in the future, but are beyond the scope of the current paper.

2. Results using the learned fear assay (Fig 5e and suppl fig 8c,d) indicate that *Trpa1*^{-/-} mice are indeed able to detect 2MT, likely via the olfactory system (Fig 5c,d). Unfortunately, no data on heterozygous or WT controls in the learned fear assay was included for comparison with the KO. This is an important control to exclude any possible reduction in olfactory performance in *Trpa1*^{-/-} mice. Reduced olfaction may have an important impact on the behavior and thus, conclusions would need to be reformulated. The cookie test shown on suppl fig 8a is clearly insufficient to evaluate olfaction in *Trpa1*^{-/-}. A habituation-deshabituation test using low 2MT dose (as on Fig 3) might be more appropriate in this case. In this context, showing that other odorants not detected by *Trpa1* (such as those used in refs 17, 47, 48) can induce freezing would greatly strengthen the work.

RE: We did not include WT or HET mice in 2MT learned fear experiment because they already showed very high level freezing in response to 2MT and were not further enhanced by pairing with electric footshock. Besides the cookie test, *Trpa1*^{+/-} and *Trpa1*^{-/-} mice showed equivalent performance in the dual odor-based learned fear assay (**Fig. 4d**), suggesting that *Trpa1*^{-/-} mice could distinguish different odors and have normal learned fear responses. As suggested by the reviewer, we performed the habituation-deshabituation test showing that *Trpa1*^{+/-} and *Trpa1*^{-/-} mice had a similar low detection threshold for 2MT (**Fig. 4c**). Taken together, these results strongly suggest that *Trpa1*^{-/-} mice have a normal sense of smell. Finally, other odorants, such as non-volatile major urinary proteins (refs 47, 48), elicit only avoidance and risk assessment behaviors, but not freezing. Rather, we synthesized and tested alarm pheromone SBT (ref 17), which is structurally related to TMT/2MT, and showed that *Trpa1*^{-/-} mice were defective for SBT-evoked innate freezing behavior (**Supplementary Fig. 5d**).

3. Whether *Trpa1*⁺ cells in the TG are driving the observed fear-like responses is unclear to me. First, the 2MT effect on behavior seems to be partial: background (no odor) freezing rates in WT mice ranges 0-20% (Fig 1a) whereas in 2MT-induced in *Trpa1* mutants is 2-fold higher, up to 20-40% (figs 2b and 3a). This would be consistent with decreased olfaction. A no odor control in *Trpa1*^{-/-} mice may clarify this point. Second, authors use a (>2s) decrease in locomotor activity as a proxy for freezing, which is assumed as a fear response. Careful observation of the movies reveals that TMT/2MT not only elicits freezing, but also sitting in distant areas from the stimulus, which can be also interpreted as avoidance. This point should be mentioned in the discussion, which now is overly oriented towards emotional fear. The “emotional” component of aversion is supposed to be lower than fear, fitting better with the classical function of TG neurons. Third, and related with the previous point, it has been reported (in ref. 13) that a subset of olfactory-activated neurons from the cortical amygdala plays a critical role in the generation of innate TMT-driven aversive behavior. Activity in this brain region in *Trpa1*^{-/-} mice in response to TMT/2MT should be reported to determine whether olfaction is dispensable for the display of the innate aversive-freezing behaviors.

RE: We include no odor control of *Trpa1*^{+/-} and *Trpa1*^{-/-} mice, both of which showed 0-20% baseline freezing rate (**Supplementary Fig. 5c**). By contrast, *Trpa1*^{-/-} mice showed 20-30% freezing rate as compare to 55-90% freezing rate of WT or *Trpa1*^{+/-} littermates in response to 2MT exposure. The results suggest that TRPA1 plays a crucial role in 2MT-evoked freezing behavior. The residual freezing response of *Trpa1*^{-/-} mice toward 2MT is probably attributed to a functional olfactory system. Moreover, we showed that *Trpa1*^{-/-} mice were also defective for the avoidance, flight, risk assessment behaviors in response to low dose 2MT (**Fig. 3d,e**) and snake skin molt (**Fig. 3f,g**). As suggested by reviewer, we include in the revision more discussion of these innate defensive behaviors in addition to freezing. Finally, we showed that 2MT evoked a similar low level of c-fos expression in the cortical amygdala (CA) of *Trpa1*^{+/-} and *Trpa1*^{-/-} mice (**Fig. 4a,b**). Importantly, habituation-deshabituation test showed a similar low detection threshold for 2MT in *Trpa1*^{+/-} and *Trpa1*^{-/-} mice. These results strongly suggest that *Trpa1*^{-/-} mice have a normal sense of smell. It should also be emphasized that we believe

both the trigeminal and olfactory systems play dual roles in innate aversive-freezing behaviors.

Minor points:

1. In fig 4 and suppl fig 6, can responses to TMT/2MT be eliminated using a specific inhibitor for Trpa1 such as HC-030031?

RE: We thank reviewer for this excellent suggestion. We showed that inhibition of TRPA1 by HC-030031 blocked 2MT-evoked Ca^{2+} transients in transfected HEK293 cells (**Supplementary Fig. 7b**).

2. Freezing rate results on *Trpv1*^{-/-} mice in response to 2MT differ considerably between fig 3b (80% approx.) and suppl fig 4f (50%).

RE: Whereas fig 3b used *Trpv1* KO mice (all males), fig 4f examined ENU mutant pedigree (males and females) that carried 40-60 mutations/exome, including *Trpv1* mutation. These ENU mutant mice showed more variable phenotypes because of two major reasons: 1) female mice generally showed less freezing response than male mice upon 2MT exposure; 2) ENU mutant mice previously endured nine different genetic screens, including a brutal chemical-induced colitis screen, which could increase the individual variations in their behaviors in our innate fear assay.

Reviewers' comments:

Reviewer #1 (Remarks to the Author):

The authors have address my concerns for the most part. I have however still a few comments:

1) The authors added a figure showing representative c-fos labeling in dorsal portion of OB for TRPA1 mutants and controls. However, without proper quantification, this figure cannot address the question of whether TRPA1 plays a role in TMT/2MT detection by dorsal OB neurons, which have been implicated in innate responses to TMT previously. This is relevant in regards to the possibility that this receptor acts not only in the TG neurons but also other input pathways driving innate fear to predator odors, which the authors discuss in the revised version of this manuscript.

Minor comments:

1) The authors should clarify the relative measure (%) of c-fos labelling in the in situs. From the graph it seems they are normalizing to the control, but it should be described clearly.

2) The authors should state in the manuscript the number of replicates regarding the experiment in figure 5d and e (cysteine mutants in HEK cells). In addition, the p values should be Bonferroni corrected. I am guessing that many of the mutation sites will survive the statistical correction.

Reviewer #2 (Remarks to the Author):

The revised manuscript incorporates new data and adequately addresses most of the initial concerns. A few minor concerns related to the description and interpretation of the data remain, which are noted below. Otherwise all comments have been addressed.

1. Author's explanation about population subsets of activated TG neurons is convincing, although robust evidence of Trpa1 as 2MT sensor in vivo in TG neurons is still missing. Therefore, I believe the authors are overstating their results in the discussion section (p. 14) claiming: "we identified TRPA1 as a chemosensor that mediates TMT/2MT-evoked innate freezing, avoidance, flight, and risk assessment". With the methods used, evidence is, at best, indirect. Authors should define it clearly.

2. New results strongly suggest that Trpa1^{-/-} mice have a normal sense of smell. Nonetheless, it cannot be completely excluded that an unidentified Trpa1-expressing subset of olfactory neurons have a partial role in the behavioral response.

3. It is surprising that Trpa1^{-/-} mice are also defective for all tested TMT, 2MT, SBT and snake skin-evoked innate fear/defensive behaviors. This may suggest a potential wiring/developmental defect rather than sensory. Partial rescue of 2MT-evoked innate freezing using AAV seems to exclude this possibility. However, there is still a possibility that overall TG function (ablated when Trpa1 is missing) is needed for any kind of freezing. In this case, Trpa1 would be a secondary housekeeping molecule for proper TG function rather than a key sensory transducer. This possibility is worth to be mention in the discussion.

Reviewer #3 (Remarks to the Author):

The authors describe a large-scale recessive genetics screen of ethylnitrosourea (ENU)-mutagenized mice designed to identify genes conferring innate fear behaviors. This reviewer was directed to comment specifically on the forward genetics screen conducted in this manuscript.

The ENU administration and downstream breeding strategy referred to in the manuscript (reference 30, Wang et al) is a well published and cited method. I have three questions about this:

1. At the bottom of page 5 the authors state that sequencing revealed on average 60 mutations/exome. Referring to the Wang et al paper, the equivalent mating schema (G0 male mated to WT female) resulted in ~45 mutations/exome
 - a. Please provide an explanation of the elevated mutation rate observed in the current study
 - b. Please state the distribution of zygosity of the mutations observed (e.g. in Wang et al, they saw on average 4 homozygous and 41 heterozygous mutations per exome)
2. The authors screened 14,390 G3 mice (487 pedigrees) for abnormal freezing response upon 2MT exposure
 - a. Please provide the statistical rationale behind this number of mice/pedigrees
 - b. Please specify if any sexual dimorphism was observed or accounted for in the screen
3. There are no details of the actual ENU administration and downstream breeding strategy included in the Online Methods section. Please correct this omission.

General comments regarding nomenclature:

1. Throughout the manuscript it looks like mouse proteins are referred to using capital letters. This is incorrect as it denotes human proteins. Mouse proteins should be written using the first letter capitalized and the subsequent letters lower case e.g. Trpa1 (Trpa1 = mouse protein; TRPA1 = human protein). Please correct throughout the manuscript.
2. Page 5 first paragraph mentions a C57BL/6 mouse. No such strain exists. Presumably the authors mean C57BL/6J. Please amend
3. Page 47 first paragraph mentions two knockout lines obtained from the Jackson Laboratory.
 - a. Please insert the full allele name then state that for brevity you will refer to them as Xxx-/- throughout the manuscript.
 - b. Note that the genetic background of the two knockout strains that you use is not pure B6J. Given the variation in phenotypes observed between different genetic backgrounds, it is critically important that you are transparent with this fact and include a statement recognizing that the genetic background of the ENU fearless mice and the imported KO strains are different.

A point-by-point response to reviewers' comments is listed below

Reviewer #1 (Remarks to the Author):

The authors have address my concerns for the most part. I have however still a few comments:

1) The authors added a figure showing representative c-fos labeling in dorsal portion of OB for TRPA1 mutants and controls. However, without proper quantification, this figure cannot address the question of whether TRPA1 plays a role in TMT/2MT detection by dorsal OB neurons, which have been implicated in innate responses to TMT previously. This is relevant in regards to the possibility that this receptor acts not only in the TG neurons but also other input pathways driving innate fear to predator odors, which the authors discuss in the revised version of this manuscript.

RE: We repeated this experiment because of poor staining quality using >1.5-year old tissue sections. Quantification of *c-fos* positive neurons showed equivalent 2MT-evoked *c-fos* activation in the dorsal region of OB in *Trpa1*^{+/-} and *Trpa1*^{-/-} mice (**Supplementary Fig. 5c,d**).

Minor comments:

1) The authors should clarify the relative measure (%) of c-fos labelling in the in situs. From the graph it seems they are normalizing to the control, but it should be described clearly.

Done (Fig. 4 legend, p23).

2) The authors should state in the manuscript the number of replicates regarding the experiment in figure 5d and e (cysteine mutants in HEK cells). In addition, the p values should be Bonferroni corrected. I am guessing that many of the mutation sites will survive the statistical correction.

Done (Fig. 5 legend, p25)

Reviewer #2 (Remarks to the Author):

The revised manuscript incorporates new data and adequately addresses most of the initial concerns. A few minor concerns related to the description and interpretation of the data remain, which are noted below. Otherwise all comments have been addressed.

1. Author's explanation about population subsets of activated TG neurons is convincing, although robust evidence of *Trpa1* as 2MT sensor in vivo in TG neurons is still missing. Therefore, I believe the authors are overstating their results in the discussion section (p. 14) claiming: "we identified TRPA1 as a chemosensor that mediates TMT/2MT-evoked innate freezing, avoidance, flight, and risk assessment". With the methods used, evidence is, at best, indirect. Authors should define it clearly.

RE: We changed this sentence (p13, line 284) to "We identified TRPA1 as a potential novel chemosensor that mediates TMT/2MT-evoked innate freezing and other fear/defensive behaviors."

2. New results strongly suggest that *Trpa1*^{-/-} mice have a normal sense of smell. Nonetheless, it cannot be completely excluded that an unidentified *Trpa1*-expressing subset of olfactory neurons have a partial role in the behavioral response.

RE: We added in the text (p14, line 302) that "We cannot exclude the possibility that an unidentified *Trpa1*-expressing subset of olfactory neurons may play a partial role in the predator odor-evoked innate fear/defensive behaviors."

3. It is surprising that *Trpa1*^{-/-} mice are also defective for all tested TMT, 2MT, SBT and snake skin-evoked innate fear/defensive behaviors. This may suggest a potential wiring/ developmental defect rather than sensory. Partial rescue of 2MT-evoked innate freezing using AAV seems to exclude this possibility. However, there is still a possibility that overall TG function (ablated when *Trpa1* is missing) is needed for any kind of freezing. In this case, *Trpa1* would be a secondary housekeeping molecule for proper TG function rather than a key sensory transducer. This possibility is worth to be mention in the discussion.

RE: We think this possibility is unlikely. It would be impossible for AAV-mediated *Trpa1* gene delivery in TG to rescue 2MT-evoked innate freezing, if *Trpa1* deficiency caused wiring/ developmental defect. Moreover, in contrast to that *Trpc2* is expressed in almost all vomeronasal neurons, *Trpa1* is expressed in only a small subset of TG neurons (~10%), and, thus, cannot function as a secondary housekeeping molecular for overall TG functions. Finally, TMT, 2MT and SBT are structural-related thiazoline odors that may function as ligands for *Trpa1*, a well-established chemosensor for pungent/irritant and potentially dangerous chemicals. We include additional discussion in the revised text (p13, lines 284-290).

Reviewer #3 (Remarks to the Author):

The authors describe a large-scale recessive genetics screen of ethylnitrosourea (ENU)-mutagenized mice designed to identify genes conferring innate fear behaviors. This reviewer was directed to comment specifically on the forward genetics screen conducted in this manuscript.

The ENU administration and downstream breeding strategy referred to in the manuscript (reference 30, Wang et al) is a well published and cited method. I have three questions about this:

1. At the bottom of page 5 the authors state that sequencing revealed on average 60 mutations/exome. Referring to the Wang et al paper, the equivalent mating schema (G0 male mated to WT female) resulted in ~45 mutations/exome

a. Please provide an explanation of the elevated mutation rate observed in the current study. RE: Please note that in the Wang et al paper, the statement is: **"...G3 descendants of a mutagenized G0 male mated to a WT female carry on average 45 mutations (~4 homozygous and 41 heterozygous) but no X-linked mutations."** This statement itself is actually in error, I think, because the expected number of homozygous mutations should be ~7.5 on assumption of phenotypic neutrality. However, it is correct that each G1 mouse has ~60 mutations, an average observed with some drift over several years of whole exome sequencing. An average of 30 mutations are thus transmitted to each G2 mouse. The average G2 mother will contribute an average of 15 mutations to each G3 pup, while the G1 will again contribute an average of 30 mutations to each G3 pup, for a total of 45 mutations in either heterozygous or homozygous form (assuming neutral transmission).

b. Please state the distribution of zygosity of the mutations observed (e.g. in Wang et al, they saw on average 4 homozygous and 41 heterozygous mutations per exome).

RE: As above, Wang et al. referred to the average G3 mouse, and actually should have been 7.5 and 37.5. In the present study, we counted the number of homozygous, heterozygous, and failed genotype mutations in every mouse. The average G3 mouse was homozygous for 5.68 +/- 3.8 mutations; heterozygous for 24.7 +/- 10.8 mutations, and indeterminate genotype for 1.01 +/- 4.2 mutations (standard deviation given). This means there were fewer mutations in the mice used in innate fear screening than in the average mice used over the whole term of mutagenesis.

2. The authors screened 14,390 G3 mice (487 pedigrees) for abnormal freezing response upon 2MT exposure

a. Please provide the statistical rationale behind this number of mice/pedigrees.

RE: Since the submission of our paper, we have realized there was a bug in our counting program, and now have amended the numbers of total mice and mutations. The new numbers do not alter our conclusions. The true total number of G3 mice was 13,222. These mice came from 632 pedigrees, which contained a total of 18,327 G3 mice (not all mice in some pedigrees were used in fear screening).

The number of mice used from each pedigree was usually maximized. Based on experience, it is extremely difficult to map a recessive mutation with fewer than 10 G3 mice. The optimal number of mice per pedigree (cost/benefit analysis) is somewhere around

40, but this number was only occasionally achieved during the study. The saturation achieved in terms of percentage of genes badly damaged or destroyed in our survey and examined twice or more in the homozygous state would be approximately 8.8% (please see Wang, T et al... Nature Communications PMID 29382827, for details of saturation computations).

b. Please specify if any sexual dimorphism was observed or accounted for in the screen.

RE: We did not observe instances of significant sexual dimorphism, but the study was not powered to efficiently detect sexual dimorphism, given the pedigree sizes we generated. Hence, such mutations may exist but may have been missed.

3. There are no details of the actual ENU administration and downstream breeding strategy included in the Online Methods section. Please correct this omission.

RE: We have corrected this omission, and thank the reviewer for his or her very thoughtful remarks.

General comments regarding nomenclature:

1. Throughout the manuscript it looks like mouse proteins are referred to using capital letters. This is incorrect as it denotes human proteins. Mouse proteins should be written using the first letter capitalized and the subsequent letters lower case e.g. Trpa1 (Trpa1 = mouse protein; TRPA1 = human protein). Please correct throughout the manuscript.

Done

2. Page 5 first paragraph mentions a C57BL/6 mouse. No such strain exists. Presumably the authors mean C57BL/6J. Please amend.

Done

3. Page 47 first paragraph mentions two knockout lines obtained from the Jackson Laboratory.

a. Please insert the full allele name then state that for brevity you will refer to them as Xxx^{-/-} throughout the manuscript.

Done. Please see the p19, lines 7-13 in Supplementary information.

b. Note that the genetic background of the two knockout strains that you use is not pure B6J. Given the variation in phenotypes observed between different genetic backgrounds, it is critically important that you are transparent with this fact and include a statement recognizing that the genetic background of the ENU fearless mice and the imported KO strains are different.

Done. Please see the p19, lines 12-13 in Supplementary information.

REVIEWERS' COMMENTS:

Reviewer #1 (Remarks to the Author):

The authors have address all my concerns.

Reviewer #3 (Remarks to the Author):

Comments from Reviewer 3 following first cycle of revision.

Thank you for addressing the issues raised. You have answered my queries satisfactorily. I have no further comments.